# Atomistic Insights of Calmodulin Gating of Complete Ion Channels

**DOI:** 10.3390/ijms21041285

**Published:** 2020-02-14

**Authors:** Eider Núñez, Arantza Muguruza-Montero, Alvaro Villarroel

**Affiliations:** Biofisika Institute (CSIC, UPV/EHU), University of the Basque Country, 48940 Leioa, Spain; enviadero@gmail.com (E.N.); arantza.muguruza.montero@gmail.com (A.M.-M.)

**Keywords:** Calmodulin, TRPV5, TRPV6, Eag1, KCNQ, SK2, SK4, Kv10, Kv7, M-current

## Abstract

Intracellular calcium is essential for many physiological processes, from neuronal signaling and exocytosis to muscle contraction and bone formation. Ca^2+^ signaling from the extracellular medium depends both on membrane potential, especially controlled by ion channels selective to K^+^, and direct permeation of this cation through specialized channels. Calmodulin (CaM), through direct binding to these proteins, participates in setting the membrane potential and the overall permeability to Ca^2+^. Over the past years many structures of complete channels in complex with CaM at near atomic resolution have been resolved. In combination with mutagenesis-function, structural information of individual domains and functional studies, different mechanisms employed by CaM to control channel gating are starting to be understood at atomic detail. Here, new insights regarding four types of tetrameric channels with six transmembrane (6TM) architecture, Eag1, SK2/SK4, TRPV5/TRPV6 and KCNQ1–5, and its regulation by CaM are described structurally. Different CaM regions, N-lobe, C-lobe and EF3/EF4-linker play prominent signaling roles in different complexes, emerging the realization of crucial non-canonical interactions between CaM and its target that are only evidenced in the full-channel structure. Different mechanisms to control gating are used, including direct and indirect mechanical actuation over the pore, allosteric control, indirect effect through lipid binding, as well as direct plugging of the pore. Although each CaM lobe engages through apparently similar alpha-helices, they do so using different docking strategies. We discuss how this allows selective action of drugs with great therapeutic potential.

## 1. Gating of 6TM Ion Channels

6TM ion channels directly modulated by calmodulin (CaM) play crucial roles in many physiological processes [1]. These channels share common architectures, but CaM regulation differs significantly. The variety of mechanisms employed by CaM testifies the amazing versatility of this protein. CaM is formed by two similar globular domains, the N- and C-lobes linked by a very flexible sequence. Each lobe is composed of two EF-hands which are responsible for binding of up to four Ca^2+^ ions. CaM targets are usually amphipathic helical protein regions rich in hydrophobic and basic residues. CaM lobes can be in an open, semi-open or closed configuration depending on Ca^2+^ occupancy. In addition, the abundance of methionine residues confers another level of plasticity at the amino acid level. These characteristics enable CaM to bind to more than 300 targets with little sequence similarity. The fact that there are three genes in humans (CALM1–3) which encode an identical CaM protein emphasizes its critical role in physiology [1]. Here, we review several high resolution structures of CaM in complex with ion channels, which provide an essential framework to understand CaM-mediated regulation. Within the channels discussed here, Ca^2+^-CaM inhibits Eag1 (Kv10) and TRPV5/6 channels, activates SK channels, and inhibits or activates Kv7 channels depending on other factors.

Among ion channels, those that selectively allow the passage K^+^ ions display the largest diversity. In the human genome, there are about 90 genes encoding different K^+^ channels, almost ten times than for Na^+^ or Ca^2+^ channels. Besides alternative splicing and RNA editing, the tetrameric combination of different subunits generates further diversity. This large variability of molecular entities is a reflection of the importance of the function “selective K^+^ permeation”, therefore, it is not surprising that it is subjected to tight regulation by several mechanisms, including the direct action of Ca^2+^. This cation is the most important inorganic second messenger, but most proteins are unable to directly interact with it, whereas CaM endows a subset of channels the capacity to respond to intracellular Ca^2+^ oscillations. The structure of three K^+^ ion channels in complex with CaM has been determined by cryo-EM at near atomic resolution, revealing unexpected features of CaM function.

K^+^ channels are enzymes that catalyze the selective passage of K^+^ ions in and out membranes through a pore down an electrochemical gradient. The mechanism involves the use of energy to remove the hydration shell around the ion. The energy is balanced when the ion recovers its hydration shell after reaching the other side of the catalytic center. Four subunits arrange around a central axis to form a symmetric catalytic center that resembles a 12 Å long tube crossing two-thirds of the membrane. The lateral chains of the signature sequence GYG or GFG, found almost invariably in every K^+^ selective channel, are pointing away from the central axis, and do not interact with the substrate. Instead, eight oxygen atoms from the backbone carbonyl groups within the catalytic tube adopt a geometry that matches with the hydration shell of K^+^ ions, but not with that of Na^+^ ions, which is key to selectivity. Crucial to the function of ion channels is gating the passage of ions, which can be interrupted at different positions along the path. A physical gate is often formed by the bundle crossing of the S6 transmembrane helix flanking the catalytic tube, which expands on activation allowing hydrated ions to flow through. The hydration of the central cavity of the pore is critical. Even the flow through large diameter pores can be prevented by subtle changes in the hydrophobicity of the surface. This feature is especially prominent for BK channels that presents a pore in the closed state as large as 10 Å in diameter, which is larger than the diameter of hydrated K^+^ (6–8 Å). However, its amphipathic surface in the closed state prevents water from passing by, becoming effectively impermeable to small ions, but allowing the passage of moderately sized hydrophobic ions such as TEA [2]. Upon activation, the channel diameter of the pore increases, but the critical feature for letting ions pass by is concealing the amphipathic Pro-Ile-Ile-Glu segment, which is achieved by a small rotation of S6 [2,3,4]. Thus, expanding of the S6 bundle is necessary, but may not be sufficient to achieve a conducting state [5]. The basic architecture of the pore is completed by an additional transmembrane segment denominated S5, as illustrated by the 2TM inward rectifier family of K^+^ channels.

Voltage-dependent K^+^ channels are clearly modular, consisting of a voltage sensing domain (VSD) located in the periphery, and a central separate catalytic pore domain (PD). The VSD consists of four transmembrane segments (S1-S4), attached to the N-terminus of the 2TM pore, acquiring a 6TM architecture. Although TRPV5/6 channels are not K^+^ selective, they will be discussed here because they have similar 6TM architecture, their gating is regulated by CaM and their 3D structures are available. In the tetramer, the VSD can be associated to the adjacent subunit in a domain swapped configuration, as in Kv7 and TRP channels discussed here, or not swapped, as for Kv10, or SK4 channels.

In some channels, like those of the Kv1 or Kv7 family, there is a region within S6 (Pro-Ala-Gly or Pro-x-Pro sequence) thought to represent a flexible hinge which allows the last portion of the S6 segment to swing away from the central axis upon activation. In channels of the Kv10–12 families that lack this hinge, introduction of a Pro at the S6 bundle results in channels that are constitutively active [6,7].

When viewed from the internal side, the coordinated bending of the four hinges leads to a counterclockwise movement, resembling the opening of an iris (see https://ars.els-cdn.com/content/image/1-s2.0-S0896627307007209-mmc3.mpg) [8]. The S6 hinge is not clearly seen in the sequence of Kv10–12 channels. But, nevertheless, a similar twist is observed in the opened Kv11 channel [9]. It is not known if the VSD is required to prevent opening at negative potentials (negative coupling) or to open the channel at positive potentials (positive coupling), and both proposals find experimental support, especially for prokaryotic channels in the first case [10], and for eukaryotic in the second [7].

Upon depolarization, the S4 segment, enriched with positively charged Arg residues, turns counterclockwise and moves outward about 10 Å (up position) [11], pulling the helix that links to S5 [8,12]. The VSD movement to the up or to the down position is coupled to the pore by several mechanisms. The most studied involves the S4/5 linker, which runs parallel to the membrane, acting as a mechanical lever. When VSD is in the up position, the S4/5 linker drags S5 to the periphery, and this movement allows the S6 bundle crossing to expand [13].

In non-domain swapped channels such as EagI and hERG (Kv10 and Kv11), S1 and S5 make extensive contacts, representing the main interface between the VSD and the PD. In these channels, the covalent linkage between S4 and S5 can be broken with little effect on the movement of S4 in response to voltage [7]. The resulting channel still gates the pore in response to voltage [14,15,16], highlighting the importance of the interface at the membrane.

## 2. Eag Channels

### 2.1. Allosteric Control of Eag Channel Gating by Calmodulin

Eag1 potassium channels (also named Kv10) belong to the KCNH or ether-a-go-go family of voltage-gated K^+^ channels and have roles in cardiac repolarization, neuronal excitability [17,18], and tumorogenesis [19]. These channels are functional when the cytoplasmic region is removed, saving the distal C-terminal coiled-coil tetramerization domain [20], or when combining split subunits, N-terminus/VSD on one side and PD/C-terminus on the other [14], highlighting the modular design. Furthermore, functional hERG channels are generated when the split point is located at the S2-S3 intracellular loop or at the S3-S4 extracellular linker [16]. Because split PD on its own is non-conducting even when carrying a mutation in S6 that generates constitutive active channels in the full-length or in the split VSD/PD configuration, it appears that PD needs to associate with VSD to adopt a configuration compatible with ion permeation [7]. It is not clear if the S6 gate is closed in the absence of VSD, or if the isolated PD does not conduct because of the collapse of the selectivity filter, the apparition of cryptic hydrophobic gates, or other reasons (see above). There is an open debate regarding the thermodynamically more stable state of the PD, open or closed. In the first case the VSD would act as an inhibitor, by actively closing the pore in the down position (de-activation by VSD down), versus the second postulate where the VSD would induce pore opening in the up configuration (activation by VSD up) [7,15,20]. 

This family of channels has an unusually long cytosolic loop joining S2 and S3. The cytoplasmic regions connected to the N- and C-termini comprise over 70% of the amino acid sequence. It includes a PAS domain (Per-ARNT-Sim) on the N terminus that is preceded by a sequence of 25–27 amino acids known as the PAS-Cap. The C-terminus, attached to the pore through the C-linker domain, contains a region with homology to cyclic-nucleotide binding domains (CNBHD) that does not bind cyclic nucleotides, followed by a coiled-coil C-terminal tetramerization domain [21].

PAS and CNBHD interact with low affinity in vitro (Kd ~13 µM), forming a complex whose structure has been solved by X-ray crystallography [22]. The crystal structure of the PAS-CNBHD complex alone deviates very little from that observed in cryo-EM images of full-channels, indicating that the transmembrane region and CaM have at best a subtle influence on the coarse architecture of the core PAS-CNBHD elements. On the other hand, significant differences are observed for PAS-Cap and C-loop [22,23], which is intriguing and may be related to the critical roles that these elements play in gating. In addition to the interactions between PAS and CNBHD seen in crystals, the images from the full channel reveal interactions of these domains with the VSD [23].

Cryo-EM images of complete channels obtained in the presence of a huge concentration of Ca^2+^ (5 mM) captured CaM at the periphery, bridging two diametrically opposed subunits [20,23] (Figure 1). The tetramerization domain, located distally in the C-terminus, was not resolved, and it is probably located in the central cavity underneath the pore gate, flanked by four PAS/CNBHD complexes. PAS and CNBHD have peripheral alpha helices, labeled N1 and C2 respectively, not present in other related channels (Kv11 or Kv12), that serve as anchoring sites for CaM. An additional helix in CNBHD, labeled C1, also interacts weakly with the C-lobe in an atypical disposition. Four CaM molecules are located in the perimeter, at 65 Å from the central axis, 40 Å from the membrane, and each CaM is far from its neighboring CaM (~90 Å). The assigned orientation for the two CaM lobes in the cryo-EM image was chosen because the proper densities were compatible with a 12.4 Å long lobe linker. In the reversed orientation, the loop would span 24.4 Å, which is impossible for a 5-residue linker [23]. The structure obtained from crystals of a peptide with the sequence of C2 grown in the presence of Ca^2+^ shows a calcified C-lobe engaging with a short helix formed by the target peptide, whereas the N-lobe is not interacting with the peptide or with Ca^2+^, which is remarkable considering the large concentration of this cation present during crystallization. The X-ray holo-C-Lobe/C2 complex obtained at 2.8 Å (PDB 5HIT) and the binding parameter between isolated CaM lobes and peptides derived from different domains [24] support the proposed orientation of the CaM lobes in the channel. The major differences between the X-ray and cryo-EM structures are that the high affinity C2 helix (PVRRLFQ) is extended by a helical turn (PVRRLFQ/RFRQQK) in the full channel, and that the presence of Ca^2+^ in the C-lobe cannot be seen on the cryo-EM structure. 

The initial attempts to localize the sites of interaction identified a region within CNBHD. Replacing Phe 714 and Phe 717 by Ser in CNBHD abolished binding to fusion proteins in vitro [25]. A peptide array screening identified two interacting sites, one at the N-terminus within the PAS domain (151–165), and another in the C-terminus (711–721), with affinities in the nanomolar range in the presence of Ca^2+^, corresponding to N1 and C2. An additional low affinity site at the C-terminus (674–683), corresponding to C1, was also identified [26]. The cryo-EM images reveal that C1 interacts weakly in an unusual way with an external surface of EF4 (Figure 2) [27]. Föster Resonance Energy Transfer (FRET) experiments suggest that this site is not able to bind CaM on its own, and that binding to either the N- or C-lobe is sufficient to engage to the channel [28]. Nevertheless, CaM is not a constitutive element, and it is coming in and out the channel at rest [25]. The cryo-EM images reveal additional unconventional interactions between external surfaces of EF3 in the C-lobe and the PAS domain of a neighboring subunit. Thus, each CaM is interacting with three subunits, two via the C-lobe and another one via the N-lobe.

Eag1 is strongly inhibited by Ca^2+^-CaM. Combining the expression of WT and CaM-insensitive subunits at different ratios, the response to Ca^2+^ of the resulting heteromeric channels is best described when only one CaM molecule out of the four present in the channel is sufficient to shut the pore. Half inhibition is observed at 70–100 nM Ca^2+^, and the suppression is complete above 150 nM [25,29], implying that under native resting conditions most of the channels should be closed (but see below). This is in line with the estimated probability of the channel being open of 0.2 at rest, indicating that almost 80% of the channels are closed at any moment [7]. In cell-free inside-out membrane patches, i.e., where the channels are completely devoid of intracellular components, the channels lose the response to Ca^2+^. The sensitivity is fully restored when the membrane patch is pushed back inside the cell or after the addition of recombinant Ca^2+^-CaM. However, only a slight increase in current density was observed, which is lower than expected if most channels were inhibited by CaM at rest and all Ca^2+^ was effectively removed. This remarkable report shows a slight increase of the current after patch excision into a medium without Ca^2+^. When the patch was bathed with 200 nM Ca^2+^ with CaM, the current was suppressed. After bathing in 0 Ca^2+^, current levels returned to the on-cell patch levels [25].

GST proteins fused to Eag1 N1 and C2 sequences bind CaM in a Ca^2+^-dependent manner, primarily reliant on Ca^2+^ occupancy at sites 3 and 4 of the CaM C-lobe. Whereas with CaM12, which abolish Ca^2+^ binding to the N-lobe, the inhibition by Ca^2+^ is marginally reduced [26]. With CaM34, which abolish Ca^2+^ binding to the C-lobe, inhibition is not present. In addition, Ca^2+^ inhibition is impaired by mutations in C2, the high affinity site for the C-lobe [25] or in channels with the CNBHD and C-linker removed (amino acids 484–668) [30] positioning the C-lobe as essential for Ca^2+^ signal transduction.

### 2.2. The PAS-Cap Domain of Eag Channels is Critical for Calmodulin-Mediated Gating

Studies with individual regions show that the PAS-Cap is structurally independent of the PAS domain, and the related domain from hERG channels in solution presents two parts, a highly dynamic 10 residues unstructured segment, followed by a stable amphipathic helix [31,32]. The structure of the PAS domain in solution is very similar to that seen by cryo-EM [9], however, none of the 20 structural NMR models for hERG reflect the position of the PAS-Cap, indicating that the interactions with other parts are crucial for proper orientation, as proposed for the related KCNH channels [15].

Earlier work reported that deleting the cytosolic N-terminal domain results in complete abolition of Ca^2+^-CaM sensitivity [26]. Latter, an increased response to voltage in a Ca^2+^-dependent-manner upon removal of the complete PAS domain was observed, i.e., in absence of the N1-site/N-lobe interaction. Remarkably, hEAG1 channels without the PAS-Cap, but with the N1-site intact, remain Ca^2+^-sensitive. Instead of being inhibited, a larger than 15-fold potentiation in current evoked is observed in response to voltage steps [30]. Whereas it is known that one calcified CaM suffice to close a WT channel, it is not yet known how many calcified CaM molecules are needed to open PAS-Cap-less channels. It is neither known how many PAS-Cap suffice to close the channel. Importantly, this potentiation requires the preservation of binding between the C-lobe and the C2 site. Thus, stabilization of the interaction of the CaM C-lobe with CNBHD facilitates opening by the VSD. This assistance is fostered by Ca^2+^-dependent stabilization of the PAS/CaM N-lobe complex. However, PAS-Cap somehow uncouples the VSD from the pore when CaM is loaded with Ca^2+^, overriding the previous mechanism and leading to gate closure. 

Thus, the PAS-Cap is essential to turn an intrinsic Ca^2+^-CaM-dependent signal that favors the adoption of an open state into a signal that leads to a non-conducting state. It is unclear how this element achieves the closure of the pore. Comparison of cryo-EM images of channels bound to Ca^2+^-CaM, with and without the PAS-Cap, reveals a rigid body clockwise rotation of the intracellular regions relative to the pore when viewed from the intracellular side (Figure 3). The movement is in the same direction as the one observed for the closely related hERG channel captured in an open state, albeit of lower magnitude. Two poses have been trapped for the PAS-Cap-less channel complexed with Ca^2+^-CaM, one with the C-terminus rotated by 2.4° and other by 8.6°, short of the 20° observed for hERG. The pore is closed, and the structures of the PAS-Cap-less channels are considered to be in a pre-open state. Thus, the VSD is uncoupled from the PD in both cases, but the role of CaM for this disengagement, if any, is unclear. 

### 2.3. Interplay between PAS-Cap and the S4/5 Linker in Eag Channels

Soon after cloning Eag1 channels, it was found that removal of PAS-Cap caused modifications in gating that were compensated by a mutation in the S4/5 linker, leading to the proposal that it may functionally interact with PAS-Cap. It was also found that the Δ7–12 deletion (RRGLVA) of PAS-Cap resulted in rapid inactivation [33]. Because the voltage-dependency was altered, the authors speculated that the N-terminus may work together with regions close to S4 within the voltage sensor. They went ahead to replace His 343 with Arg, because *Drosophila* Eag differs in voltage-dependence and has an Arg at this position. Although in rat Eag1 the presence of this mutation has only minor effects, remarkably, it compensated the effect of Δ7–12. In a noteworthy study, it was found that breaking the channels in two at position Leu 341 resulted in constitutive activity, and replacing Asp 342 for varied amino acids, except Asn, greatly facilitated closure, leading to the proposal that Asp 342 participates in an interaction that favors the open state [7]. The possibly that the side chain participates as a hydrogen bond acceptor was pointed out because among many replacement tested, only the isosteric Asn could mimic Asp. Latter, a detailed mutagenesis-function study confirmed the critical role of Asp 342 in gating [20]. This presumed interaction has not been resolved in the available cryo-EM structures [20,23]. 

The functional compensation of Δ7–12 by a S4/5 mutation, the proximity of the PAS-Cap in the open conformation of hERG [33], and the impact of Δ3–9 and R7A/R8A mutant on function, suggest that R7/R8 might form a functional interaction with D342 in S4/5 [20]. Mutation-function studies suggest that PAS-Cap can be divided into two segments, the unstructured N-terminus (residues 1–9, not seen in the cryo-EM structures) and the unstructured C-terminal (residues 10–13) that precedes the amphipathic alpha helix (residues 16–25). The N-terminus seems to promote opening, whereas the C-terminus appears to promote closure [20]. 

### 2.4. The PAS/CNBHD Complex of Eag Channels is not Compacted by Calmodulin

The distance between the two main CaM anchoring residues, Trp 148 in PAS with the N-lobe and Phe 714 in CNBHD with the C-lobe, is about 42 Å. This is similar to the longest separation previously determined for the Holo-CaM Munc13 complex, illustrating the amazing gymnastics that CaM is able to achieve [34,35]. Apparently, CaM is clamping PAS towards CNBHD. However, the crystal structure obtained without CaM overlaps almost perfectly (RMSD 0.504 Å) with that of the channel engaged with Ca^2+^-CaM and trapped in a pre-open state. Thus, compaction of the PAS/CNBHD complex *per se* does not appear to underlie CaM-mediated Eag1 regulation. Furthermore, superposition of the cytoplasmic region, with or without PAS-Cap, does not reveal any noticeable structural difference [22,23]. Thus, how binding to CaM at the periphery permeates ~80 Å to the PAS-Cap to promote closing the channel is unclear [20].

## 3. SK Channels

### 3.1. The N-Lobe Acts upon the S4/5 Linker and the ViSD to Mechanically Pull the Gate Open of SK4 Channels

In 1996 a sequence with homology to the pore region of K^+^ channels was used to probe rat and human cDNA libraries, leading to the identification of genes encoding Ca^2+^ activated K^+^ channels with 550–600 residues [36,37]. Few years later, CaM was identified as the accessory element conferring Ca^2+^ sensitivity [38,39]. Voltage-insensitive CaM activated K^+^ channels K_Ca_2.1 to K_Ca_2.3 are collectively known as small-conductance SK channels. K_Ca_3.1 channels (SK4) are also known as intermediate conductance IK channels, and are approximately 40% identical to other SK channels. SK channels contribute to the after-hyperpolarization following an action potential and mediate the intrinsic excitability of many cells [40], modulate the activation of immune responses [41], and contribute to the regulation of vascular tone [42].

These 6TM channels share ~40% residue conservation with voltage-activated K^+^ channels. Although the S4 segment contains three positively charged Arg residues, opening is insensitive to transmembrane voltage [43]. The voltage insensitive ViSD adopts a non-swapped arrangement similar to Eag channels, and interacts with the PD of the same subunit. The basis for the position of ViSD relative to the pore is unclear because, in contrast to Eag channels, the S4/5 linker is relatively long. The two first TM segments, S1 and S2, are unusually long, entering deep into the cytosol, such that the intracellular base can interact with CaM in a non-canonical manner (Figure 4 and Figure 5). 

The intracellular region immediately following S6 is conserved, especially the initial segment, which plays a critical role in transmitting Ca^2+^ signals to the pore. The CaM binding domain (CaMBD) is formed by two antiparallel alpha helices (hA and hB) that run parallel to the plasma membrane, resembling the C-linker of Eag channels, but extending much further to the periphery, exceeding the diameter defined by the transmembrane domains (Figure 4). The CaMBD is followed by helix C that tends to run perpendicular to the membrane and contributes to the tetramerization coiled-coil domain that follows. This segment is not entirely defined in cryo-EM images because of its flexibility [44]. Mutagenesis experiments suggest that the proximal helix C region influences how CaM transmits Ca^2+^ signals to the gate [43].

At resting Ca^2+^ levels, CaM is bound to the closed channel and subsequent Ca^2+^ elevation leads to pore opening. Combining the expression of WT and CaM-insensitive subunits at different ratios, the response to Ca^2+^ of the resulting heteromeric channels is best described if the concerted action of all four molecules is required for gating, which readily provides an explanation for the steeply cooperativity with Hill coefficients of 3–4 [39]. In contrast to Eag1, the response in excised patches (devoid of intracellular components) persists, and CaM remains bound to the channels even in the absence of Ca^2+^. An additional difference is that the response to Ca^2+^ is barely affected by the C-lobe incompetent CaM mutant CaM34, but is precluded by CaM12. Thus, Ca^2+^ binding to the N-lobe triggers channel opening [39]. 

CaM is constitutively bound through the C-lobe to the distal part of helix A that follows transmembrane S6. This interaction alters the geometry of the C-lobe EF hands, such that one or both are incapable of binding Ca^2+^ [44,45], highlighting one more time the complexity of the relation between Ca^2+^, CaM and target. Besides its role in Ca^2+^ signaling, CaM is critical for surface expression and tetramerization [46,47]. 

A series of remarkable coincidences led to a striking model for Ca^2+^-CaM-dependency channel activation. First, a Ca^2+^-independent binding site was identified in the distal part of helix A, whereas a site whose interaction with CaM absolutely required Ca^2+^ was mapped to the distal C-tail (in helix C) [39]. Second, the structure of the C-tail/CaM complex solved by X ray crystallography revealed a dimer in which CaM was hugging two CaMBD, establishing in 2001 a new record for the separation of the two main target anchoring residues at the N- and C lobes at 25 Å, a score that pales with the actual mark set in 2010 at about 42 Å for the Munc/CaM complex (see above). The structure revealed an antiparallel disposition of helix A and helix B-C. Helix B is followed by helix C, but this helix is disguised in the X-ray structure, appearing as a continuation of helix B, precluding its recognition as a different element. Helices B-C in the dimer adopt an antiparallel hB-hB’/hC-hC’ disposition, with the Ca^2+^-loaded N-lobe engaging (artefactually) helix C’ from the other subunit. Consistent with bridging, the apparent binding affinity of the CaMBD for CaM is reduced two fold in the presence of Ca^2+^, from 8.2 to 3.7 nM [48]. Third, the functional effect of crosslinking agents in mutant channels with strategically engineered cysteine residues could be rationalized within the X-ray structure, suggesting that the artefactual interaction between Glu 404 (Glu 295 in SK4) and Lys 77 in the CaM lobe linker seen in the X-ray structure existed in the full-length channel [49]. Fourth, drugs that enhance SK2 function were co-crystallized with the CaMBD making contacts with the proximal part of helix C, and the effects of mutations in helix C on drug affinity was somehow rationalized within the X-ray structure [50,51].

The model proposed that CaM bridges two subunits together when the N-lobe binds Ca^2+^, and then the channel adopts a dimer of dimers configuration, causing the rotation of S6 and pore opening [44]. The idea was directly challenged in 2014, because no evidence was found for the existence of 2:2 complexes, and it was proposed that CaM was crosslinking adjacent subunits [35,52], but it prevailed until it was definitely discarded in 2018 based on cryo-EM images [45].

### 3.2. The N-lobe Mediates Gating on SK Channels

A common approach to interrogate molecular mechanics is to introduce reactive Cys at key positions, and compare the function before and after treatment with thiol-reactive methanosulfonate reagents (MTS). It was concluded that there is no impediment for ion flow at the cytosolic end of S6, because residues Ala 283 to Ala 286 are entirely exposed to water in the open state, and MTS reagents can reach A283C and A286C mutated residues in closed channels [53]. Further studies led to the proposal that the gate of SK4 could be described by a narrow passage centered at Val 282 (Val 390 in SK2) [54], closer to the selectivity filter than in other channels. Satisfactorily, this conclusion based solely in careful functional studies was confirmed in the structure of the full channel [45].

The notable analysis of cryo-EM images of the SK4/CaM complex has provided astonishing detailed information on gating [45]. In the absence of Ca^2+^, the C-lobe is fixed to helix A, whereas the N-lobe is mobile (Figure 4). From time to time, the N-lobe lands into one of the four cavities formed around the long S4/5 linkers and flanked by the proximal part of helix C and the base of the S1 segment of two neighboring subunits. The N-lobe has been trapped at three preferred positions, from S1 at the periphery to the S4/5 linker in the middle of the ViSD. No structural PDB files for these three structures are available. In the movie based on this data, a rigid body movement of the N-lobe is observed, always with the EF hands closed, with no discernible interactions within the intracellular pocket. The C-lobe remains static during these N-lobe excursions (https://science.sciencemag.org/content/suppl/2018/05/02/360.6388.508.DC1). The N-lobe was poorly solved in the absence of Ca^2+^, and only channels with just one site occupied of the four were captured, suggesting that the interactions between apo-N-lobe and the cavity are very weak and transient [45]. 

The S4/5a helix, conserved among SK channels (with only one residue difference out of 10), is the target for Ca^2+^-N-lobe (Figure 5), where Leu 185 serves as the main anchoring residue, and Ile 182 is the secondary anchoring site. Modifications with thiol-reactive reagents in channels with an engineered Cys at this helix prevented activation by Ca^2+^, reinforcing the proposed role of S4/5a as the genuine CaM target in SK channels. Replacing Leu 185 for Ala surprisingly has little consequences for Ca^2+^ sensitivity (EC_50_ 0.44 µM for WT and 0.40 µM for the S181A/L185A double mutant) [54]. In contrast, replacing Ser 181 with Trp or Tyr results in channels that produce large currents at low Ca^2+^ concentration, which is consistent with improved docking of the mutant S4/5a into the N-lobe [55]. However, when interpreting these data it has to be considered that mutations at sites than cannot interact with CaM can produce even larger effects in Ca^2+^ responsiveness (see below). This is because sensitivity depends on at least two processes: binding the N-lobe to its effector site, followed by mechanical coupling to the gate (see for example [56,57]).

The N-lobe engages helix S4/5a in a forward orientation, whereas there is a backward orientation in “artefactual” SK2/N-lobe X-ray structures [44,58]. Holo-N-lobe is also interacting with the cytosolic intrusion of the unusually long S1 transmembrane segment through the proximal part of the lobe linker. On the other side, the EF1-EF2 linker makes contacts and runs parallel to the proximal part of helix C of a neighbor subunit. The N-lobe slides up flanked by S1 and hC from two subunits, and traps S4/5a when loaded with Ca^2+^, while the C-lobe maintains its interaction with helix B. The N-lobe then pulls S4/5a downward and this displacement expands the S6 bundle crossing and opens the pore [45]. Because helices AB run parallel to the membrane and crosses underneath the pore axis to the other side, the C-lobe interacts with a subunit diametrically opposed to the one engaged by the N-lobe (Figure 6). This is well suited for a coordinated action of CaM on the four subunits, readily explaining cooperativity and the negative dominant effect of CaM-insensitive variants [39]. 

Pore opening is linked to a rigid body rotation of the helix AB/C-lobe complex around an axis defined by Arg 316 (Asn 426 in SK2) in helix A. The comparison of the closed and open states reveals that Arg 316 overlaps almost perfectly in both structures. The distal part of the AB/C-lobe complex rises towards the membrane, whereas the proximal region swings in the opposite direction, changing the angle between helix B and helix C. The AB lever then pushes S6 centrifugally, dilating the pore to an open configuration. Additionally, the proximal region of helix A interacts with the S4/5a helix from an adjacent subunit, which may explain the increased Ca^2+^ sensitivity due to mutations to bulky hydrophobic residues at this region (Val 407 and Met 411 in SK2, equivalent to Val 298 and Met 302 in SK4) [56].

Close inspection of the N-lobe reveals that the groove between the expanded EF1 and EF2 hands is only partially filled by the S4/5a helix. The main interaction takes place between Leu 185 in S4/5a of SK4 and Leu 39 in helix 4 of CaM EF2. Within the lobe linker of CaM, Met 76 establishes contacts with Leu 12 and Arg 15 at the base of S1, and simultaneously with Arg 180 at the beginning of S4/5a. Thus, through interactions within the lobe linker and EF2, the N-lobe simultaneously inclines the ViSD and pulls down S4/5a. On the partner subunit, the anchorage with the C-lobe is reinforced through new contacts between Glu 86, at the other extreme of the CaM lobe linker, and Arg 355 in the middle of helix A of SK4 [45]. Mutations at the proximal region of helix A or in the S6-helix A linker, alter Ca^2+^ sensitivity [49,56,59,60], which fit with a plausible enhancement/interference of the transmission of a pulling force to the gate. In addition, it is necessary to move simultaneously S5 away to accommodate the S6 bundle crossing expansion. This is fulfilled by pushing the cytosolic base of S1-S3 towards the periphery and pulling down S4/5a. These movements tilt the ViSD from an adjacent subunit and, in turn, drag S5 away from the central axis, allowing the gate to expand. In summary, a coordinated movement of the ViSD and PD of all subunits takes place upon N-lobe engagement with S4/5a and S1.

### 3.3. Some Clinically Relevant Drugs Stabilize the Interaction with the N-lobe in SK Channels

Several compounds that enhance SK function have been co-crystallized with the isolated CaMBD of SK2 [50,51]. Importantly, riluzole, the first medication approved for amyotrophic lateral sclerosis is among them [60,61,62]. The docking site is located at the N-lobe. Although these structures artefactually place helix C engaging the N-lobe, these drugs can fit into a binding pocket defined in the native structure (Figure 7), where the N-lobe is occupied by S4/5a [55]. The X-ray structures showed that EBIO, the more potent NS309 and some derivates bind at the interface between the N-lobe and helix C [49,50,59]. A series of recent mutagenesis-activity and modeling studies are consistent with the idea that the S4/5a shapes the interaction interface for these drugs [55]. The major contribution to the binding energy are Van der Waals forces with the largely hydrophobic pocket in the N-lobe. Remarkably, the same residues in the CaM N-lobe appear to contribute to drug binding in the isolated CaMBD and in the full-channel [55]. In fact, a computer based screening using the artefactual binding structure identified 30 hit compounds, and two of them (SKS11 and SKS14), that closely resembled 1-EBIO and NS309 in their chemical structure, potentiated the current. The difference in the contribution to the binding energy by the proximal helix C and the S4/5a helix may be small, because the potency of these four compounds correlated wonderfully with the interaction energy computed from the artefactually crystal structures [59]. Modeling studies suggest that the drug binding site can be reshaped by mutations in the proximal helix C that interacts with the EF1-EF2 linker, influencing the engagement of the N-lobe with S4/5a [55]. 

The pharmacology of the CaM site demonstrates that CaM can be a selective site of action for some drugs. This selectivity arises from the diverse orientation that different targets adopt when anchored to CaM (Figure 7). For instance, the orientation of the docked helix within the N-lobe for the channels discussed here varies significantly, making this pocket a promising target for drug development.

### 3.4. The Interaction between SK Channels and Calmodulin is Regulated by CK2-Mediated Phosphorylation of the Lobe Linker 

Protein Casein kinase 2 (CK2) and protein phosphatase 2A (PP2A) form a signaling complex with SK2 channels and regulate channel activity through phosphorylation/dephospohylation [63]. CK2 docks to an N-terminal binding site (125-RRALF-129 in SK2, equivalent to 15-RKRKL-20 in SK4) located at the base of S1. CK2 requires positively charged compounds to phosphorylate CaM [64], but these compounds are not present in functional excised patch experiments. As an alternative, it has been suggested that the charged residues at the base of S1 serve this purpose [63]. It has been shown that the mechanism involves phosphorylation of Thr 79 at the CaM lobe linker, leading to an increased dependency on PIP_2_ binding, which is a lipid co-factor required for the function of SK2 channels [51]. The expression of a phosphorylated CaM surrogate (CaM T79D) reduces Ca^2+^ sensitivity of SK2 channels. CaM phosphorylation occurs only when the channels are closed [63], which nicely fits with the observation that under resting conditions the N-lobe is detached most of the time from the channel and, therefore, the lobe linker is exposed. Molecular docking of PIP_2_ to the X-ray structure led to the proposal that the S6-hA linker (K397, K402, K405 in SK2, corresponding to K288, K293 and K405 in SK4) contributed to the binding site in conjunction with R74 and R77 in the CaM lobe linker [65]. However, the cryo-EM structure of SK4 is not compatible with such shared interface for PIP_2_ binding [45]. Contrary to this proposal, T79 is not located in the vicinity of the putative binding site constituted by the S6-hA linker. Nevertheless, the positions of the positively charged residues within the channel are sound for an interaction with a membrane lipid. It is also conceivable that the CaM N-linker contributes to a different PIP_2_ interface shared with other regions of the channel, especially with a stretch of positively charged residues at the base of S1 (13-RRRKR-17), which are at reasonable distance. In particular, Arg 15 in S1, Arg 180 at the beginning of S4/5a, and Arg 191 at the beginning of S4/5b point towards the CaM lobe linker, and are well suited to conform a PIP_2_ binding site which can be affected by phosphorylation at CaM T79, or by mutations at R74 or R77. This potential existence of such a site suggested by the cryo-EM structure could explain the increased dependency on PIP_2_ observed for the phosphomimetic T79D CaM mutant [51]. As mentioned above, in the activated channel, Met 76 at the CaM lobe linker establishes contacts with Arg 15 at the base of S1, and thus, PIP_2_ could “glue” the CaM linker to the base of S1 to promote the displacement of the ViSD during activation (Figure 8).

## 4. TRPV5/6 Channels

### 4.1. A Ball and Chain Mechanism for Holo-CaM-Dependent Inactivation of TRPV5 and TRPV6 Channels

TRPV5 and TRPV6 are close homologs (~75% sequence identity) that belong to the transient receptor (TRP) family of ion channels, but contrary to other family members, they are not thermosensitive or activated by ligands. Both are expressed in epithelial cells, TRPV5 mainly in the kidney and placenta, whereas TRPV6 in intestine [66]. They are characterized by a high selectivity for Ca^2+^ over monovalent cations (P_Ca_/P_Na_ ≥100), and play critical roles for Ca^2+^ homeostasis [67,68,69]. The high Ca^2+^ selectivity is determined by a single aspartate residue (TRPV5-D542, TRPV6-D541) in the reentrant loop at the pore [70]. These channels are tightly regulated by PIP_2_ and CaM. While PIP_2_ stimulates, CaM inhibits activity, preventing excessive Ca^2+^ influx [71,72]. Small changes in intracellular Ca^2+^ prevent the entry of this cation via these channels because half maximal inhibition is observed at about 90 nM [67]. 

Removal of the distal C-terminal of TRPV5 (S698X) or mutations at W702 and R706 in the C-terminal tail abolishes the sensitivity for CaM, resulting in enhanced Ca^2+^ flow [72,73]. Similar effects are also observed on TRPV6 mutants [74], where a double mutation in the distal C-terminal CaM binding site (W695A-R699E TRPV6) essentially eliminates inhibition by CaM in excised patches [75]. Pull down experiments suggest Ca^2+^-dependent CaM binding also to the N-terminal tail, and, more weakly, to the transmembrane region of TRPV6 [71].

Reminiscent of Eag1 channels, in excised membrane patches lacking intracellular components, TRPV6 channels do not respond to Ca^2+^. Subsequent addition of CaM recovers Ca^2+^-dependent current suppression, meaning that CaM is not constitutively bound to the channel [75]. Thus, CaM is entering and exiting the channel constantly. The number of channels engaging CaM depends on the availability of this protein that differs wildly from the total concentration (eg. 10 µM total, 100 nM free). In many cells, the number of targets exceeds that of CaM, so CaM availability becomes limiting and is generally unknown [35,76]. In fact, a significant reduction of TRPV6 activity and increase in current inactivation are observed when CaM is overexpressed [74].

TRPV5/6 channels are composed of four subunits, each containing a 4TM domain, similar to a VSD but insensitive to voltage (ViSD), a PD domain in a swapped configuration, with intracellular amino and carboxi termini, resembling 6TM potassium channels (Figure 9) [66]. The presence at the N-terminus of a number of repeating ankyrin domains having a canonical helix-loop-helix fold is a defining feature [66]. Classically, ankyrin repeat proteins are built from tandems of two or more repeats and form curved solenoid structures that are associated with protein-protein interactions. At the C-terminus, the S6 segment is followed by a hallmark alpha helix, denominated TRP helix, involved in channel gating [77], which runs parallel to the membrane underneath the ViSD up to the base of pre-S1. Following the TRP helix, the post-TRP segment forms an extended loop that engages with the N-terminal intracellular alpha helix. This helix runs almost perpendicular to the membrane and parallel to the ankyrin repeats of an adjacent subunit and with the pre-S1 of its own subunit, clamping adjacent subunits together [70].

Similar to other TRP and K^+^ channels, the inner gate is formed by a S6 bundle crossing (Figure 9). A mutational screening of residues homologous to other TRP channels located at positions near the S6 bundle crossing constriction led to the identification of Trp 583 as a component of the inner gate [78]. This residue is conserved among different TRPV5/6 species, but not in other members of the TRPV family. In a homology model built using a TRPV1 structure as template, W583 was positioned at the intracellular end of S6. The theoretical analysis of the different Trp rotamers at this position suggested a wide range of gate diameters (11.2 to 3.1 Å) depending upon the orientation of this residue. In the narrowest configuration, the pore diameter was estimated to be ~0.8 Å, which is smaller than the diameter of dehydrated Ca^2+^ (0.99 Å). Together with the altered function of W583 mutants, it led the authors to the insightful proposal that this residue serves as a channel gate [78]. This assignment has been confirmed in a series of full-channel structures [79,80,81]. The intracellular ankyrin repeats at the N-terminus are placed at the periphery, creating, in conjunction with the base of the membrane domains, a spacious dome underneath the central pore. Towards the membrane, there are four large windows ~24 Å in diameter, flanked by the base of the transmembrane domains at the top (from a lateral view), the pre-S1 and post-TRP domains on the upper laterals and the ankyrin repeats at the lower sides. 

Three groups have solved structures of TRPV5 or TRPV6 channels by cryo-EM with CaM engaged within the dome [79,80,81], with resolutions between 3.3 to 4.4 Å, which provided detailed information of CaM-mediated inactivation. Remarkably, all the structures, from different isoforms and species, are very similar (TRPV5 and TRPV6 RMSD < 1.5 Å). All these structures show one CaM molecule blocking the pore in a manner not seen before. The position of the linker that joins EF hands 3 and 4 is the key for inactivation [79,80,81]. Lysine 115, at the center of this linker, sticks out its side chain into the open channel gate formed by the four tryptophan 583 residues of each channel subunit. This makes the surface of interaction with Lys 115 the largest of any CaM interacting residue with the channel (Figure 10). This kind of interaction, with four tryptophan residues forming a cage around the ammonium group of lysine, has not been observed in any other protein structure. Interactions with residues downstream of Trp 583 also contribute to the binding site, where the second most important residue is Gln 587 for TRPV5 and His 587 for TRPV6 (Figure 10).

In a sense, this resembles the ball and chain mechanism of inactivation of some voltage-dependent K^+^ channels (Figure 11): it represents the obstruction of an open channel by a peptide [82]. Similar to K^+^ channels, there are four “ball and chains” per channel, and only one suffices to block the ion path [79,80,81]. It differs in the localization of the blocking site. For TRPV5/6 it is positioned in the lower mouth of the pore, whereas for K^+^ channels the site is believed to be at the inner vestibule, just after the selectivity filter [83]. Following with this analogy, the “ball” corresponds to the EF3/4 linker, and the “chain” could be assimilated to the complex between CaM and the C-terminal tail [84]. The C-tail is not seen in cryo-EM images in the absence of CaM, indicating the flexibility of this segment. Even in the presence of large concentrations of CaM and Ca^2+^, only fragments of the C-tail are resolved.

The “chain” is composed of two alpha helices, C1 and C2, where the N- and C-lobes engage, respectively [79,80,81]. C1 and C2 are separated by a ~40 residue flexible linker, not seen in the cryo-EM images. CaM adopts a stretched pose, where each lobe target is independent of the other. The distance between the main anchoring sites (F651 in C1 and W701 in C2) is ~29 Å. Helix C2 is trapped in the V-shaped groove of the C-lobe at the base of the pore with an orientation similar to that adopted by the high affinity site in Eag1 channels (also denoted C2, but with different amino acid sequence), but it lands in a backward orientation [79,80,81]. Whereas the C-lobe backbone architectures when engaged to Eag1, SK4 or TRPV5/6 are indistinguishable, the orientation of the target helices within this V-groove differs significantly among them [79,80,81], highlighting one more time the versatility of CaM for target recognition.

There are interactions with other regions that contribute significantly to the contact surface (Figure 10) that can be divided in four parts. Besides the critical interaction with the pore, there are contacts dispersed over the ankyrin repeats that, taken together, cover ~916 Å^2^, surpassing the contact area between the C-lobe and helix C2 (~782 Å^2^), but lower than the surface between the N-lobe and helix C1 (~1162 Å^2^). Overall, the contact surface with the N-lobe is ~2078 Å^2^, almost triplicating that of the C-lobe with C2. Within CaM, the largest interacting elements are h1 and h2 (EF1), h6, h7 and the h6–7 linker (EF3/4 linker) [79,80,81].

### 4.2. Variable Stoichiometry of the Calmodulin-TRPV5/6 Complexes

Most particles in the cryo-EM studies are trapped with just one CaM molecule. In the structure with the highest resolution, additional CaM molecules were identified within the dome [81]. In 35% of the particles, just one CaM molecule could be identified. In >20%, another slightly weaker density corresponding to the N-lobe bound to an opposite subunit was discerned, whereas in a small number of cases, this second density corresponded to the N-lobe bound to an adjacent site. A weak signal assigned to the second C-lobe was also seen in a number of particles, which suggests that the C-lobe is very mobile when not hooked to the pore [81]. In the TRPV6 structures, a unique feature was observed: helix C1 is partially formed in the adjacent subunit to the one occupied with the N-lobe, but not engaged with another N-lobe [80]. Instead, it seems to be stabilized by unconventional interactions with CaM docked to the neighboring C1 helix. This helix is not resolved in the remaining two subunits of the tetramer. It seems that a site is being prepared for docking another CaM, poised for a rotary movement as CaM molecules come in and out the dome.

There is a differential functional effect of over-expression of CaM Ca^2+^-insensitive mutants (CaM1234 or CaM34) in TRPV5/6 channels. These CaM mutants reduce significantly current density of TRPV6 but not that of TRPV5 channels. In fact, CaM34 increased the current density of TRPV5. In contrast, over-expression of CaM12 did not significantly affect TRPV5/6 activity [71]. The effect of CaM1234 is contradictory, because another study reported that Ca^2+^ influx thought TRPV6 was unaffected by this CaM mutant, and no signs of interaction with the channel were revealed by FRET [74]. In an effort to understand this differential behavior, chimeric channels were constructed. Exchanging the N and/or C termini for TRPV6 by that of TRPV5 did not fully prevent the CaM34 induced reduction in activity, leaving the transmembrane domain of TRPV6 as the common element of all channels that are sensitive to CaM34. The authors suggested that the dominant-negative effect of CaM34 is due to constant anchoring of CaM to TRPV6, but FRET studies did not support this hypothesis [74]. These findings are perplexing taking into account the structural similarities between TRPV5 and TRPV6, and suggest slight differences in the way CaM interacts with each channel family. 

Whereas the final stage in the inactivation process is evident, the steps leading to inhibition are open to speculation. The importance of W583 is obvious in the cryo-EM images, and supported by the functional consequences of certain mutants, because replacement with Leu generates Ca^2+^-CaM-insensitive channels [79]. It is generally assumed that the C-lobe engages to C2 before the N-lobe grabs C1 because the affinity for Ca^2+^ is six fold larger than for the N-lobe [81]. However, Ca^2+^ affinity changes dramatically in the presence of targets, and the N-lobe affinity can surpass that of the C-lobe in some circumstances [85]. The lack of effect of over-expression CaM12 suggests that Ca^2+^ binding to the N-lobe is not a pre-requisite for inactivation [71]. Two densities corresponding to the N-lobe where identified in a significant number of particles, whereas a second C-lobe was barely visible, and a pre-formed N-lobe target helix was clearly visible in TRPV6 particles [79,80,81]. Thus, there might be conditions within the dome that make the binding to the N-lobe more favorable. Furthermore, the contact surface with the N-lobe is significantly larger than that of the C-lobe. Thus, the role of each CaM lobe and the sequence of events remains an open question. 

## 5. KCNQ channels

### 5.1. KCNQ Channels Can be Potentiated or Inhibited by Calcium 

KCNQ2–5 channels, mainly expressed in the nerve system, are the molecular components of the M-current, and KCNQ1, expressed mostly in cardiac tissue combined with accessory subunits, constitute IKs [86]. The M-current takes its name because activation of muscarinic receptors suppressed a K^+^ conductance in sympathetic neurons causing an increase in excitability. It was soon found that the second messenger cascade invariably involved activation of phospholipase C, with the associated production of IP_3_ and release of Ca^2+^ from intracellular stores [87]. Therefore, Ca^2+^ became a suspect, and some early work pointed at this second messenger as the main culprit for current inhibition [88,89]. However, more than 20 years after the discovery of the M-current, the “mystery second messenger” was finally revealed [90,91]. It turned out to be PIP_2_, which is a co-factor absolutely required for KCNQ channel function. When PIP_2_ levels drop, the VSD disengages from the PD, and the channel cannot be opened in response to voltage changes [92]. 

The effect of Ca^2+^ on M-current is complex. Modest increases enhance the current in sympathetic neurons, and suppression is seen at higher concentrations [93]. M-channel activity is reversibly suppressed by Ca^2+^ in excised inside-out patches of sympathetic neurons, indicating that kinases or phosphatases are not mediating the effect [94]. The rise of intracellular Ca^2+^ and CaM stimulation due to activation of the phospholipase C cascade in *Xenopus* oocytes expressing KCNQ2/3 heteromers causes an enhancement of the M-current, whereas Eag1 currents are completely suppressed [95]. Thus, Ca^2+^ can cause both inhibition and potentiation of neuronal KCNQ currents. In contrast, only potentiation of the cardiac IKs or KCNQ1 currents has been described [96,97]. It is generally assumed that CaM is directly involved in Ca^2+^-dependent regulation of KCNQ channels, although only indirect evidences are available.

KCNQ channels present a VSD swapped 6TM architecture, with a long C-terminal tail that harbors five alpha helices [98,99]. Helices A and B are topologically equivalent to the C-linker of Eag1 or helices A/ B of SK channels. Similarly, they adopt an antiparallel fork configuration and tend to run under the VSD. Whereas in Eag1 and SK channels helices A/B run rather parallel to the membrane, A/B has been trapped at about 45° to the membrane in KCNQ1 channels, leaving room so CaM can be positioned between the AB fork and the VSD (Figure 12). Helix TW (or post-helix A), named this way due to the similarity to the SK2 C-lobe docking helix sequence, is located after helix A in a long and poorly conserved linker [100]. There is a sharp turn following helix B, so helix C goes perpendicular to the membrane forming a loose bundle between the four subunits, in line with the tendency of helix C to form weak dimers in vitro [101]. A flexible sequence with no conservation among KCNQ subunits follows to link with helix D that forms a coiled-coil tetramer, a feature also observed in Eag1 and SK channels [21]. The global architecture is notably similar to that of SK channels, with the CaM C-lobe also engaging helix A almost with the same orientation, even at the lateral chain level [35]. However, the C-lobe lands into the proximal part close to the gate in KCNQ channels, the helix A/B fork is much shorter, and helices A and B are connected by a much longer flexible loop. Departing from the S-shaped CaM structures discussed previously, CaM adopts a more compact C-shaped configuration, embracing the A/B fork, where helix A pairs with the C-lobe and helix B marry the N-lobe [48]. 

Similar to SK channels, trafficking of KCNQ channels to the membrane requires CaM [57,96,102,103,104,105], and current density is affected by CaM availability: increases, decreases, or no effects upon CaM overexpression have been reported [106,107,108,109,110]. Since the CaMBD has a notorious aggregation tendency when produced in bacteria in the absence of CaM, it is thought that CaM is a constitutive auxiliary subunit of KCNQ channels. Nevertheless, some mutant channels compromised for CaM binding are fully functional, suggesting that KCNQ constitutive CaM binding, or resident CaM, is not a strict requirement for function [95,111], and that CaM may exit and return to the channel under some circumstances (see below).

The structure of the full-channel with Ca^2+^-loaded CaM has been captured in a configuration in which S4 is in the up position, but the VSD is disengaged from the PD. It has been assumed that this non-functional configuration represents the end stage adopted when plasma membrane PIP_2_ levels drop [98]. In the cryo-EM particles and derived structures, four compact domains can be identified: the VSD, the PD, the CaMBD, and the distal tetrameric helix D coiled-coil. The helix D coiled-coil tetrameric domain is clearly seen, but no attempt to derive an atomistic model of this region has been reported, because of its inherent flexibility. For the same reason, the long linker between helix A and B is not seen in the atomistic model, and therefore it is not known if helices A and B come from the same or different subunits. The PD is in a closed configuration, very similar to that of other closed structures of potassium channels. 

### 5.2. The PIP_2_ Site Delineated by CaM and Helices B/C Linker is Far from the Membrane in the Cryo-EM Structure of KCNQ1

Mutations at diverse positions, generally positively charged Arg and Lys, have profound effects on PIP_2_ regulation, defining at least two positively charged pockets [98]. These residues are clustered in different regions of the channel, at the base of S1, the S2/3 linker, the S4/5 linker, the S6-helix A linker and at the helix B-C linker regions [98,112,113]. Resembling SK channels, the CaM lobe linker (Arg 74 and Lys 75) participates, in conjunction with the helix B-C linker, in shaping one of these PIP_2_ binding sites, which was further characterized by mutagenesis and molecular dynamics using a crystallographic KCNQ1 CaMBD structure [114]. The position of this particular PIP_2_ binding site is, however, ~10 Å away from the membrane in the collapsed cryo-EM structure (Figure 13). Thus, in addition to the VSD, in the absence of PIP_2_ the CaMBD apparently is displaced from its functional position. To determine the location of the CaMBD in functional KCNQ channels, the effect of hooking a pore blocker to CaM has been evaluated [115]. The K^+^ channel blocker TEA was attached to CaM with different tether lengths, and injected into *Xenopus* oocytes where translation of KCNQ2/KCNQ3 subunits took place. The length of the tether that led to 50% current reduction relative to the maximum was determined to ascertain a likely distance from the pore. The functional triangulation placed CaM ~10 Å closer to the cytoplasmic gate than the cryo-EM derived structure. Two CaM residues (Thr 45 and Thr 110) displayed a similar tether length-current density relationship, placing them at a comparable distance (~50 Å) from the pore, whereas in the cryo-EM images those residues are located at ~60 Å. Intuitively, it can be envisioned a rigid body displacement of the CaMBD towards the membrane to match the distances of this presumed PIP_2_ site and CaM. However, the third coordinate obtained with the CaM-TEA-tethering approach positioned Thr 35 at ~40 Å from the pore, whereas this residue is about 70 Å away in the cryo-EM structure. Satisfying these distance constrains may require an additional rigid body rotation of the CaMBD. Thus, given these discrepancies, it is advisable to exert caution when interpreting the KCNQ1 structure and functional trigonometry data available. In any event, the take-home message is that there is a ~10 Å incertitude regarding the relative position of the CaMBD. 

### 5.3. Calmodulin may Act Directly upon the S6 Gate of KCNQ Channels 

Several structures of the isolated KCNQ CaMBD have been solved with different number of Ca^2+^ sites occupied (PDBs: KCNQ1: 4V0C, 4UMO; KCNQ2: 6FEG, 6FEH; KCNQ2/3 chimera: 5J03; KCNQ4: 6B8L, 6B8M, 6B8N, 6B8P, 6N5W; KCNQ5: 6B8Q). All of them show a similar overall architecture with helix A engaging the C-lobe, and helix B docking into the N-lobe, matching the preferences for binding underscored in vitro [48,116]. The first complex was crystallized without adding Ca^2+^, but the N-lobe sites where occupied whereas the C-lobe where not, suggesting that the affinity for Ca^2+^ in the N-lobe increases when it is complexed with helix B of KCNQ channels, surpassing that of the C-lobe [117]. The structure obtained was a dimer, reminiscent of the crystal SK2 dimer. The relevance of this dimeric arrangement was examined in a series of clever concatameric constructs, reaching the conclusion that the dimeric arrangement is not relevant and that CaM is in a *cis* position (helices A and B from the same subunit) in the tetrameric channel. However, when mutant channels impaired in CaM binding and function due to mutations at either helix A or B are combined, functional activity is recovered [118], indicating that it is possible for CaM to embrace different subunits. Nevertheless, this *trans* configuration is believed to represent a minority [119].

There are contrasting results regarding the effect of Ca^2+^ on the affinities of the different components. There is agreement in that the affinity between helix B for the N-lobe is higher than that of helix A for the C-lobe [110,118]. The relative strength of interaction between these components changes in opposing ways, and the N-lobe is more efficient at displacing CaM from the complex at low Ca^2+^, whereas in the presence of this cation, the C-lobe becomes more efficient [48]. Experiments done at very low concentrations to avoid the strong tendency of helix A and B to aggregate, consistently show that the affinities for the different components are weaker in the presence of Ca^2+^ [120]. In contrast, when higher concentrations are employed, no binding or very weak interactions are detected at low Ca^2+^ levels by isothermal calorimetry (ITC) or microscale electrophoresis (MTS) [85,110]. The comparison of the structure with and without Ca^2+^ at the C-lobe shows that the number of hydrogen bonds increases for the N-lobe by 7% and decreases for the C-lobe by 12%, whereas the number of hydrophobic contacts diminish by 7% and 31% for N- and C-lobe, respectively, and direct contacts are reduced by 10% for both lobes [121], which is in line with the weakening of the complex caused by Ca^2+^ observed for KCNQ1 and KCNQ2 complexes [48,97,100]. 

Structural rearrangements resolved by NMR of the CaMBD in KCNQ2 channels produced by Ca^2+^ show that the N-lobe/helix B complex remains almost unchanged, whereas the initial part of helix A bents by about 18° concomitant to a torsion of the C-lobe around the helix A axis [121]. This proximal helix A segment is attached to S6, close to the bundle crossing (Figure 14). It is very suggestive that the movement caused by Ca^2+^ is poised to affect the expansion of the S6 bundle crossing seen in a model for the KCNQ2 activated state [122]. The relatively minor movement of the proximal helix A cannot account for the overall increase in volume of the CaMBD complexes caused by Ca^2+^ in vitro or at low free CaM levels in vivo [85,108,123]. It has been proposed that this volume increase could be due to detachment of the C-lobe from helix A or even helix A unwinding [85], but FRET or NMR fails to detect such rearrangements [121]. On the other hand, a number of structural models displaying varying volumes are compatible with the structural restrictions imposed by NMR. In these models, the TW helix is close to helices A/B in the most compact complexes, whereas a more extended pose, in which the TW helix tends to become a continuation of helix A is associated with a larger volume [121]. Interestingly, mutations in this element, some found in human disease, have a strong impact in both CaM binding and channel function [100]. 

### 5.4. Calmodulin May Alter PIP_2_ Binding to KCNQ Channels by Two Mechanisms

In addition of the presumed direct coupling with the pore doorway, CaM calcification influences gating by affecting PIP_2_ binding in a subunit-dependent manner [109], and there is reciprocal Ca^2+^-dependent competition for binding to the site in helix B described above between PIP_2_ and the N-lobe [97]. The dynamics of the interaction between the N-lobe and the CaMBD are enigmatic due to indications that, at rest, the N-lobe is already calcified, and, therefore, all of the Ca^2+^ sensing should be mediated by the C-lobe. However, phosphomimetic mutations at helix B can compromise CaM binding [99], suggesting a dynamic CaM binding scenario governed by PKC activity [124]. Indeed, an alternative view in which the N-lobe plays a more prominent role in signaling has been proposed. This model in which the N-lobe binds first to helix B, requires the existence of channels devoid of CaM, i.e., CaM not playing a mandatory structural role [110]. 

Both CaM and PIP_2_ dependent regulation are affected by the tetramerization domain. A pathological mutation that weakens helix D coiled-coil debilitates the interaction with CaM and generates channels that are less sensitive to PIP_2_ depletion [108]. How this allosteric influence is transmitted from an element located that far from the membrane is a mystery, and the structure of the full-channel does not provide clear clues [119].

Besides the interactions with helices A and B seen in the crystallographic and NMR structures, the cryo-EM images reveal additional potential interactions between the EF3 Ca^2+^ binding loop and the S2/3 linker (Figure 15 and Figure 16), which is unusually long among voltage-dependent K^+^ channels (with the exception of Eag1 channels). Interestingly, this linker may participate in PIP_2_-mediated coupling between the VSD and the PD [112]. In the 3D cryo-EM image, this linker, together with the base of S1 and the initial part of the S4/5 linker, constitutes a second positively charged pocket adequate for PIP_2_ binding [98], and the interaction between EF3, S1 and S2/3 suggests another mechanism by which CaM can interfere with PIP_2_ binding (Figure 16). The EF3 loop is distorted, and presumably is unable to bind Ca^2+^ due to the interaction with the base of the VSD. Shifts on voltage-dependency observed upon over-expression of CaM and CaM mutants have been related to this interaction [85] (but see [110]), suggesting that Ca^2+^ can also affect gating by a direct interaction between CaM and the voltage sensor.

### 5.5. The Interaction between KCNQ2 Channels and Calmodulin is Regulated by CK2-Mediated Phosphorylation of the Lobe Linker 

Another remarkable similarity with SK2 channels is that the interaction between CaM and KCNQ2 channels is regulated by phosphorylation of the lobe linker [125]. CaM binding to the channel is improved by phosphorylation, which makes the channels more resistant to PIP_2_ depletion [109,126]. Protein phosphatase 1 (PP1) binds to the N-terminus of KCNQ2 and, in conjunction with CK2, controls CaM phosphorylation status [127,128], and, consequently, how tight CaM is attached to the channel. In contrast to SK2 channels [125], changes in Ca^2+^ sensitivity were not detected [126]. Another difference is that SK2 anchors PP2A instead of PP1 [125,126]. Again, the impact of this phosphatase/kinase tandem implies that CaM may be entering and leaving the channel dynamically, rather than being a fixed structural constituent of the channel. Mutagenesis experiments identified three potential sites of phosphorylation (Thr 79, Ser 81 and Ser 101), being, like for SK2 channels, Thr 79 the most critical of the three [126]. 

## 6. Summary

This table summarizes the main properties of the regulation by CaM of the channels reviewed.


**Eag1****SK****TRPV5/6****KCNQ**CaM DockingN-lobeN1 (holo)[23,24]S4-S5 (holo)[45]C1 (holo)[79,80,81]hB (holo)[117]C-lobeC1, C2 (holo)[23]hA (apo)[45]C2 (holo)[79,80,81]hA (apo)[117]Ca^2+^ signaling
C-lobe[30]N-lobe[45]EF3/4 linker[79,80,81]C-lobe[85,97,121]Residence/motilityN-lobeDynamic[25]Dynamic[45]Dynamic[75]Static[85,97,121]C-lobeDynamic[25]Static[45]Dynamic[75]Partially static[85,97,121]Mechanism
Allosteric(S4/5, PAS cap)[20]Mechanical(ViSD, S4/5a)[45]Direct pore[75]Indirect(PIP_2_, S6 gate)[97,121]EC_50_ Ca^2+^
70–100 nM[25,29]440 nM[50]90 nM[67]?Ca^2+^ Effect
Inhibition[25,29]Activation[39]Blockade[79,80,81]?

## 7. Outlook

The molecular understanding of the varied conformations that CaM adopts when regulating different targets provides a rich ground for the development of tailored therapies. It is conceivable a portfolio of molecules acting upon CaM complexes, yet having exquisite selectivity and specific therapeutical applications. The cryo-EM revolution has increased the pace at which structural information of membrane proteins is obtained and has provided a better understanding, although still partial, on how CaM regulates ion channel function. The wealth of structural information is providing a strong framework for setting theoretical grounds for the underlying mechanics of critical processes. This explosion on new information has highlighted the need to comprehend how a constellation of atomic coordinates change with time in response to varying environments. The actual empirical era will move towards a new one based on the application of basic physical principles. We expect a new revolution in biology by extending molecular dynamics simulations to longer time scales through novel knowledge-based algorithms and acceleration of computer processing, perhaps on the hands of quantum computing.

## Figures and Tables

**Figure 1 ijms-21-01285-f001:**
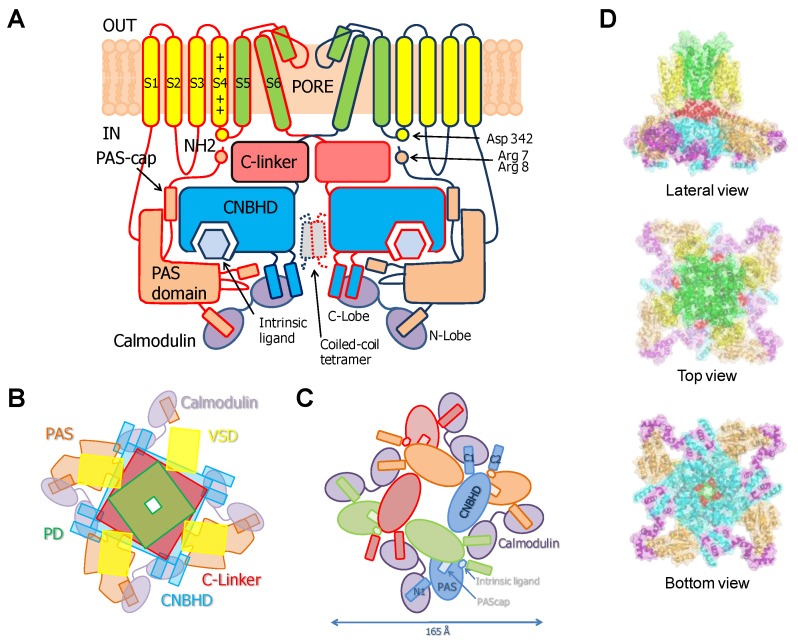
Representation of an Eag1 channel (**A**). Transmembrane regions are in yellow for the Voltage Sensitive Domain (VSD), and green for the Pore Domain (PD). Only two of the four subunits are represented, with a red or blue border. The N-terminal PAS-Cap and Pas domain in orange, the C-linker in red, the CNBHD in cyan, and CaM in purple. The critical Asp 342 in S4/5 and Arg 7 and 8 are highlighted. The presumed position of the tetrameric coiled-coil is indicated with dashed lines. (**B**). Realistic representation of the channel views from the top using the same color scheme as in A. (**C**). Realistic representation of a bottom view, excluding the C-linker. Each subunit has a different color. CaM bridges two diametrically opposed subunits. (**D**) Structures visualized using then PDB coordinates (6PBY), rendered with Pymol 1.30 using the same color scheme as in panel A.

**Figure 2 ijms-21-01285-f002:**
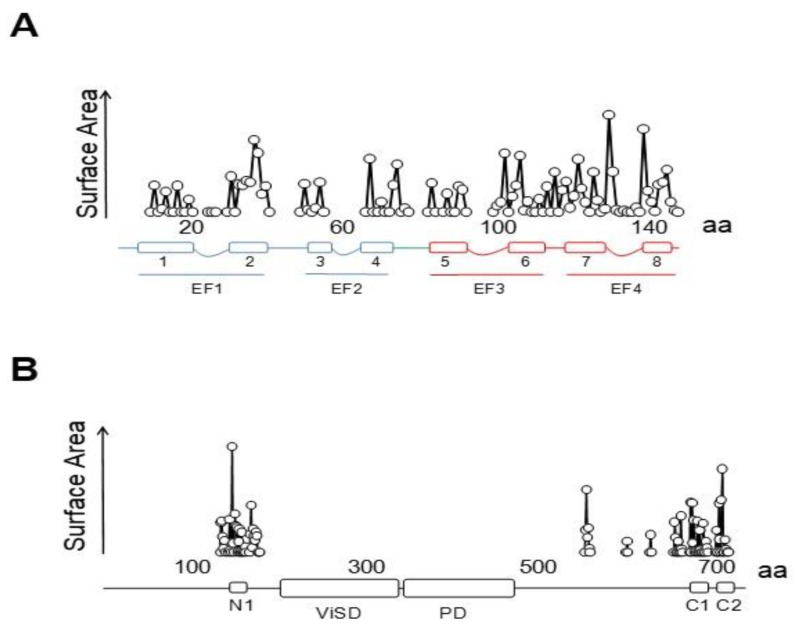
Plot of the contact surface [27] per residue in CaM (**A**) and Eag1 (**B**). Based on PDB 5K7L, 6PBX and 6PBY. “Contact surface area” between atoms A and B is defined as the area of the sphere whose center is the center of atom A and whose radius equals the sum of the Van der Waals radii or atom A and a solvent molecule. The solvent-accessible and contact surface of every atom has been computed using the software made available by Sobolev et al., http://oca.weizmann.ac.il/oca-bin/lpccsu [27].

**Figure 3 ijms-21-01285-f003:**
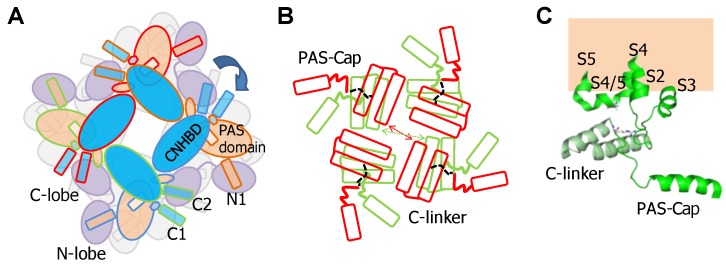
Movements during gating of Eag channels. (**A**). Realistic representation of the rigid body torsion of the intracellular cytoplasmic based on the structure of the closed Eag1 channel and the opened hERG channel, viewed from the bottom. (**B**). Realistic representation of the relative motion of the C-linker (joined rectangles representing antiparallel alpha helices) and the PAS-Cap. The dashed black line represents the presumed position of the N-terminus not seen in the cryo-EM images. (**C**). Structure of the C-linker and PAS-cap from opened hERG (PDB 5VA1). Arg 4 (corresponding to Arg 7) and Asp 540 (corresponding to Asp 342) are represented as stick using CPU coloring scheme. The orange box indicates the presumed position of the border of the membrane.

**Figure 4 ijms-21-01285-f004:**
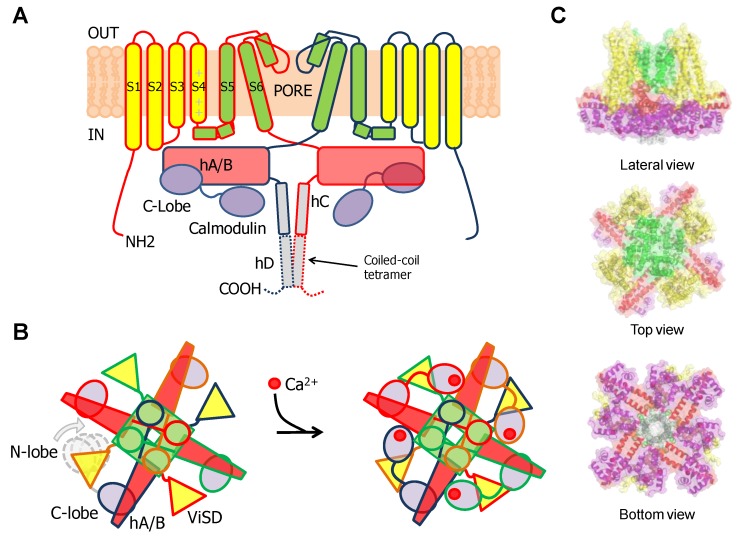
Representation of SK channels. (**A**). Transmembrane regions are in yellow for the Voltage inSensitive Domain (ViSD), green for the PD and calmodulin in purple. Helices A and B are colored in red and run parallel to the membrane under the diametrically placed subunit. The presumed position of the tetrameric coiled-coil is indicated with dashed lines. Only two of the four subunits are represented, with a red or blue border. (**B**). Realistic representation of the channel view from the top using the same color scheme as in A. Each subunit has a different color. On the left, under resting conditions the N-lobe (dashes lines) wanders around and occasionally enters into the space underneath the subunit next to the C-lobe docking site, with three preferred positions. On the right, upon Ca^2+^ loading (red circle) the N-lobe docks into and makes contact with three different subunits. (**C**). Structures visualized using then PDB coordinates (6CNO), rendered with Pymol 1.30 using the same color scheme as in panel A.

**Figure 5 ijms-21-01285-f005:**
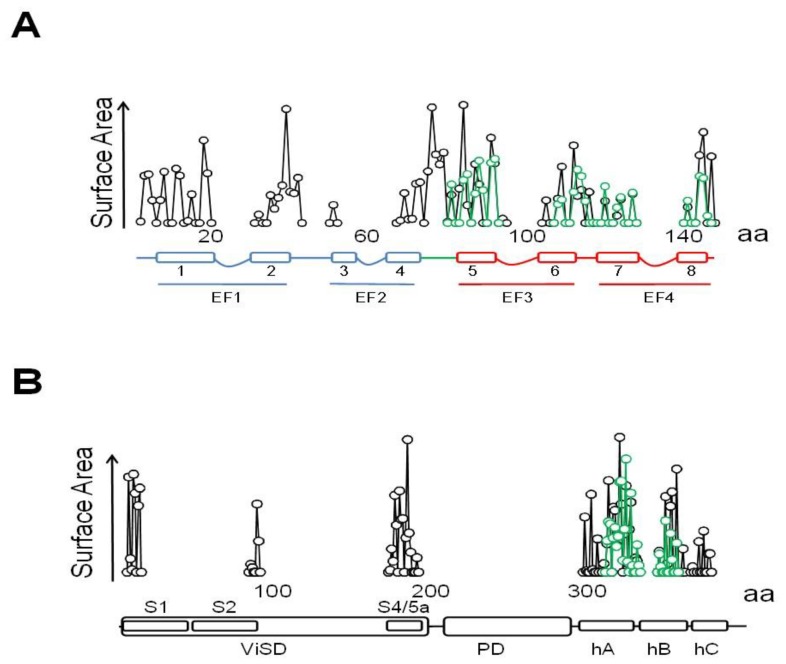
Plot of the contact surface per residue in CaM (A) and SK4 (B) in apo (green) and loaded with Ca^2+^ (black). (Based on PDB 6CNM, 6CNN and 6CNO).

**Figure 6 ijms-21-01285-f006:**
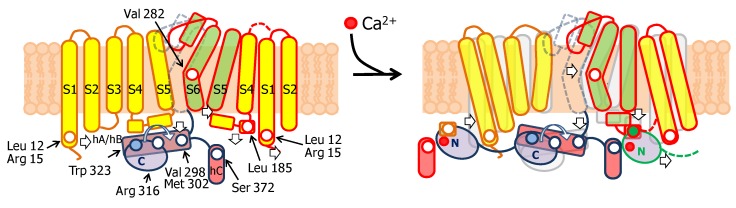
Cartoon representation of the coordinated movements upon SK4 activation. Same coloring scheme as in Figure 4. Part of the PD of the blue subunit is represented with dashed lines. Note that the complete ViSD of the red delimited subunit has not been represented to simplify the scheme. Also, S3 of the red delimited subunit, as well as the ViSD and S6 of the blue subunit have been left out. The main CaM anchoring residues Leu 185 in helix A and Trp 323 in helix B are filled with the color of its corresponding subunit when engaged with CaM. The remaining residues contact CaM as indicated in the main text, represent the axis or helix A/B rotation (Arg 316), or constitute the pore gate (Val 282). The border of CaM is colored according to the subunit engaged with the C-lobe. Note that the N-lobe of blue CaM interacts with helix C of a red subunit, and S1 of an orange subunit.

**Figure 7 ijms-21-01285-f007:**
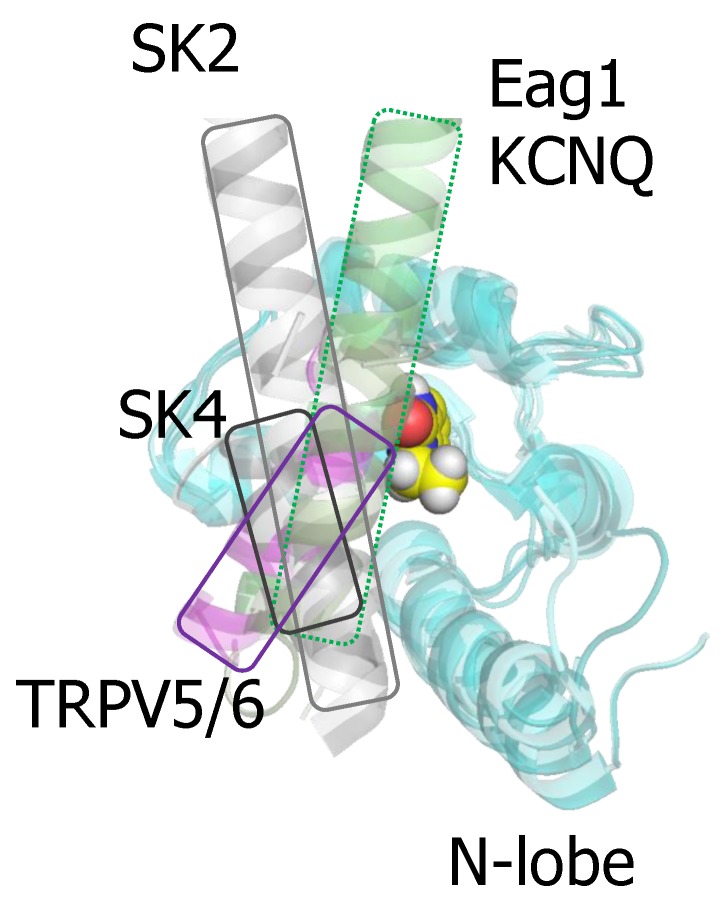
The N-lobe drug binding site of SK channels. Superposition of the N-lobe engaged with the target helix of TRPV5/6 (purple), SK4 (dark grey), SK2 (grey), Eag1 and KCNQ (green) on the complex of SK2 with EBIO (PDB 4G28). The position of the helices from Eag1 and KCNQ do not appear to be compatible with the presence of this drug.

**Figure 8 ijms-21-01285-f008:**
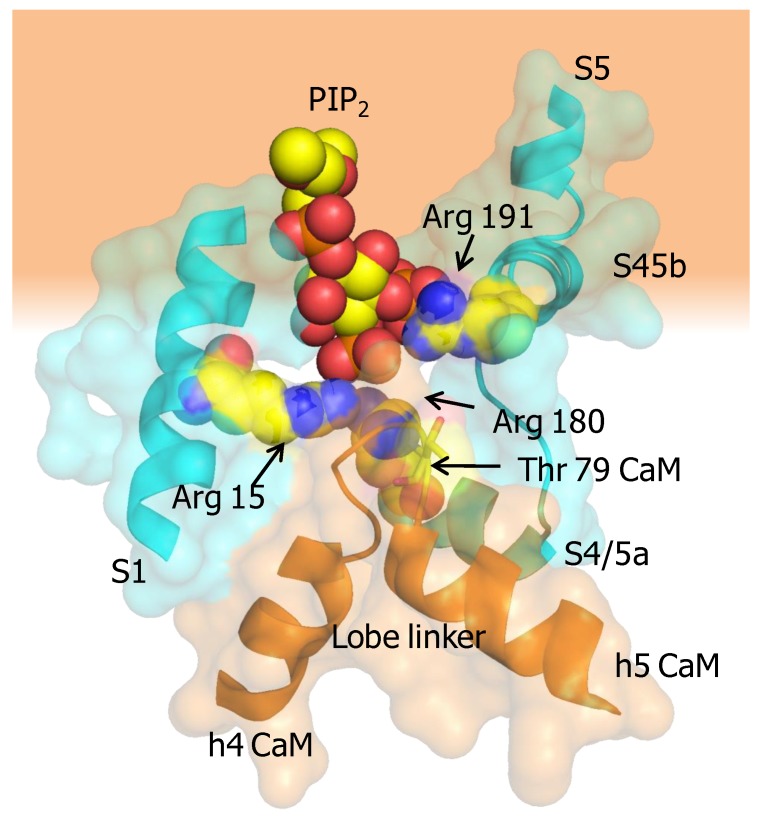
Hypothetical PIP_2_ binding site near the CaM Thr 79 phosphorylation site. The pore residues (GYG) were used as a reference to align the Kir structure (PDB 3SYA) to the open configuration of SK4 (PDB 6CNO). The PIP_2_ molecule was manually moved maintaining the distance from the membrane to the presumed docking site, and manually adjusted to avoid obvious clashes. The docking space is delimited by S1, S2 and S4 transmembrane segment. The positively charged Arg residues are represented as spheres using the CPK coloring scheme, and CaM Thr 79 is represented as stick.

**Figure 9 ijms-21-01285-f009:**
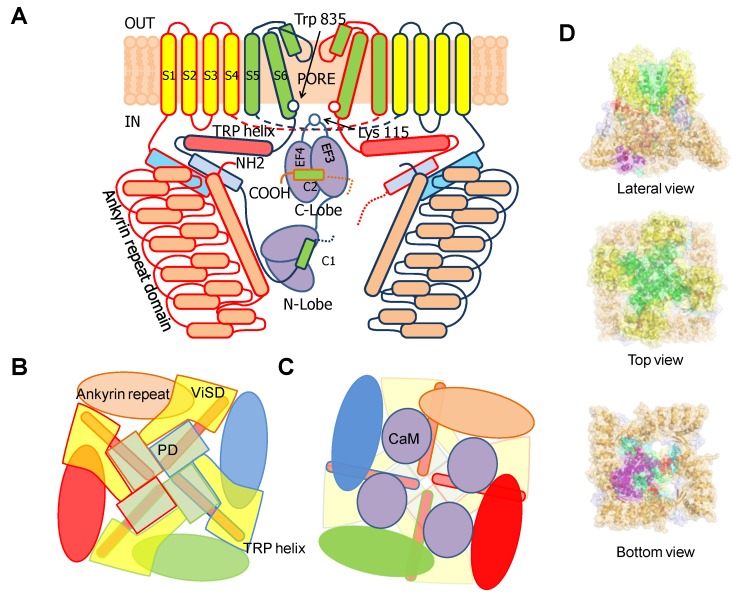
Representation of TRPV5/6 channels. (**A**). Transmembrane regions are in yellow for the ViSD, green for the PD and calmodulin in purple. Only two of the four subunits are represented, with a red or blue border. The ankyrin repeats are in orange, and the TRP helix in red. (**B)**. Realistic representation of the channel view from the top using the same color scheme as in A. The border of each subunit has a different border color and the ankyrin repeats have been filled with its corresponding color. (**C**). View from the bottom. Four hemi-CaM molecules within the dome are represented. (**D**). Structures visualized using then PDB coordinates (6E2G), rendered with Pymol 1.30 using the same color scheme as in panel A.

**Figure 10 ijms-21-01285-f010:**
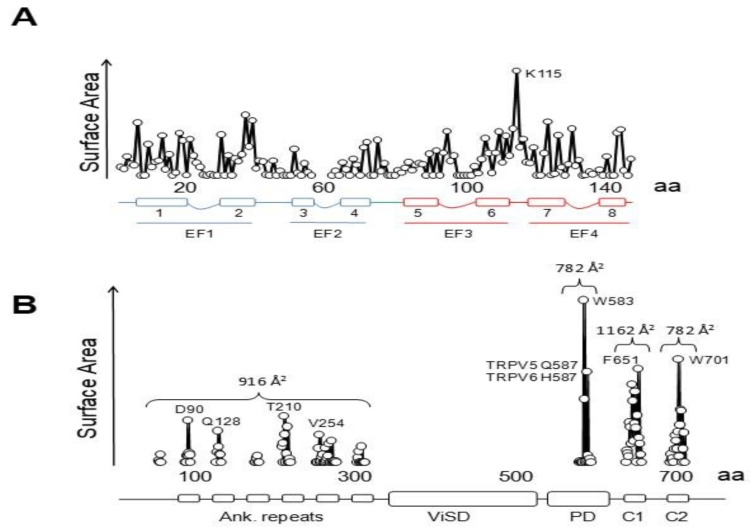
Plot of the contact surface per residue in CaM (A) and TRPV5/6 (B). Based on PDB 6DMW, 6O20, 6E2F.

**Figure 11 ijms-21-01285-f011:**
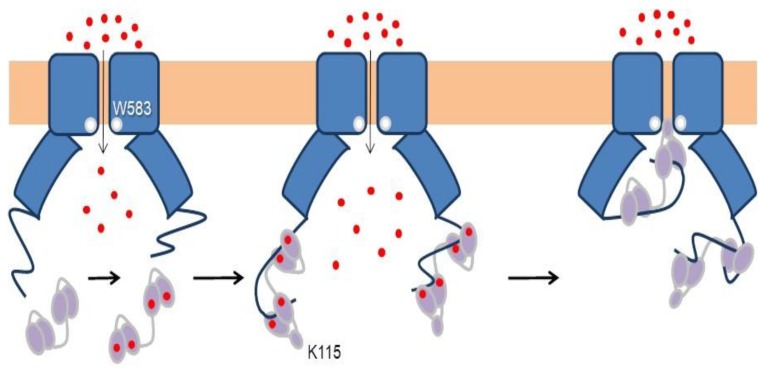
Ball and chain-like mechanism of TRPV5/6 channel inactivation. The influx of Ca^2+^ triggers CaM binding to the C-terminal tail. CaM docks within the intracellular dome in a position that favors plugging the pore with Lys 115 located in the middle of the EF3/4 linker. The ensuing decrease on Ca^2+^ within the dome should cause the release of CaM, followed by the re-initiation of another cycle.

**Figure 12 ijms-21-01285-f012:**
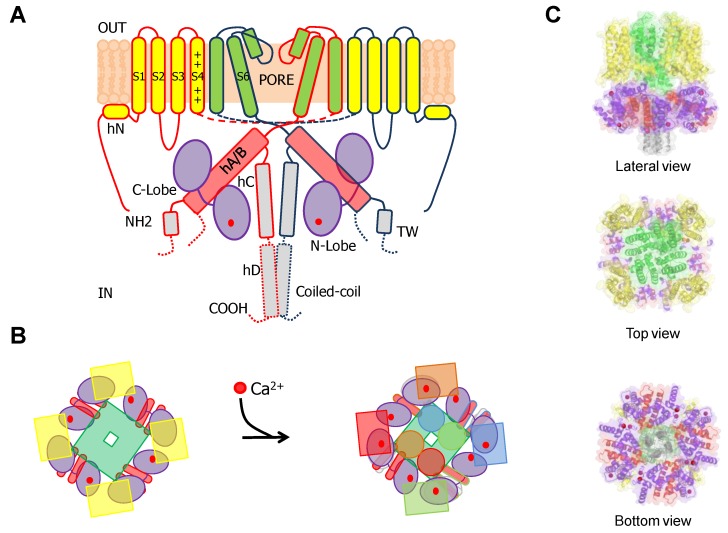
Representation of a KCNQ channel. (**A**). Transmembrane regions are in yellow for the VSD, green for the PD and CaM is in purple. The cytosolic helix parallel to the membrane before S1 is labeled as hN. Helices A and B, colored in red, are at ~45° to the membrane under the VSD from the same of subunit. The border is colored red or blue to identify each subunit. To account for swapping, the VSD and PD of two different subunits are represented continuously. The presumed position of the tetrameric coiled-coil is indicated with dashed lines. (**B**). Realistic representation of the channel view from the top using the same color scheme as in A. Helices A and B are represented in red. On the right, upon Ca^2+^ loading (red dots) there is a small movement of helix A. Each subunit has a different color in this cartoon. (**C**). Structures visualized using then PDB coordinates (5VMS), rendered with Pymol 1.30 using the same color scheme as in panel A.

**Figure 13 ijms-21-01285-f013:**
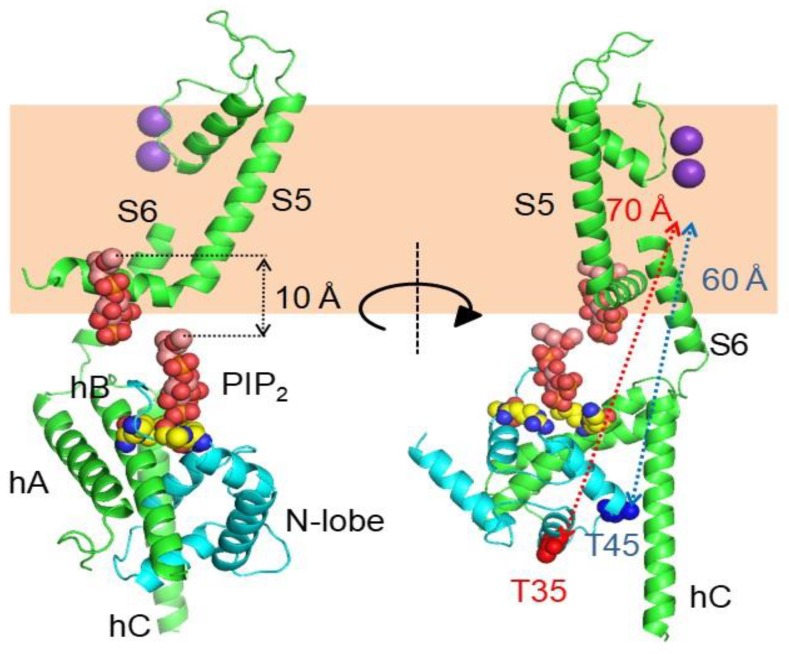
Two views of hypothetical docking of PIP_2_ with the channel at the site in which CaM participates. The pore residues (GYG) were used as a reference to align the Kir structure (PDB 3SYA) to KCNQ1 (PDB 5VMS), placing a reference PIP_2_ molecule adjacent to the S4/5 linker. The second PIP_2_ molecule was placed manually at the CaM/helix B-C linker site. Only the N-lobe of CaM is shown (cyan). On the right, the distances for T35 and T45 to the putative inner TEA binding site are indicated. The purple balls represent K^+^ ions. The orange box represents the possible location of the membrane.

**Figure 14 ijms-21-01285-f014:**
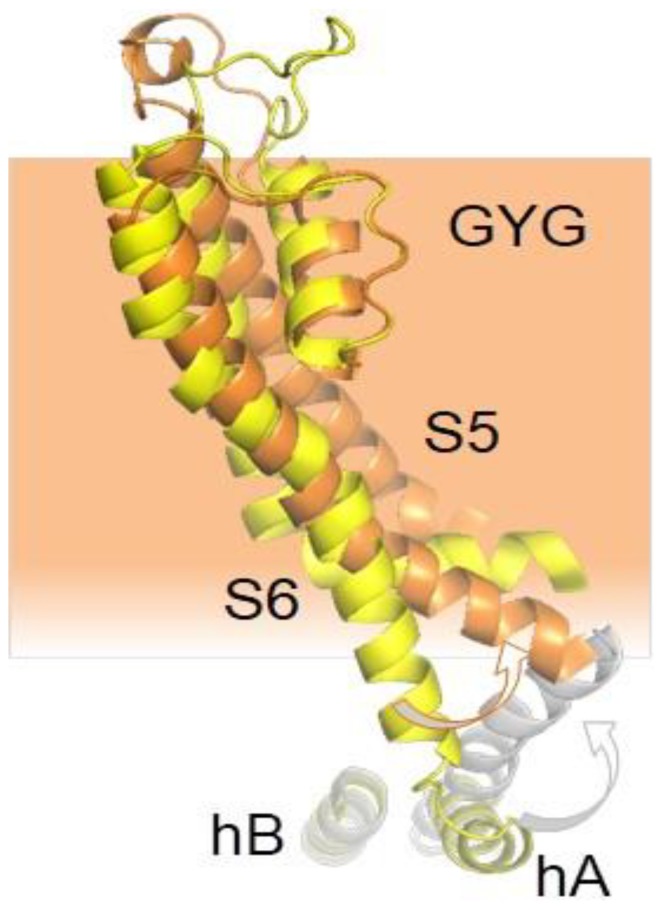
Possible direct gating transmitted from helix A to S6 in KCNQ2 channels. Comparison of the movement of S6 from a model (orange [122]) of KCNQ2 opening with the position of helix A upon C-lobe Ca^2+^ loading (grey, PDB 6FEH).

**Figure 15 ijms-21-01285-f015:**
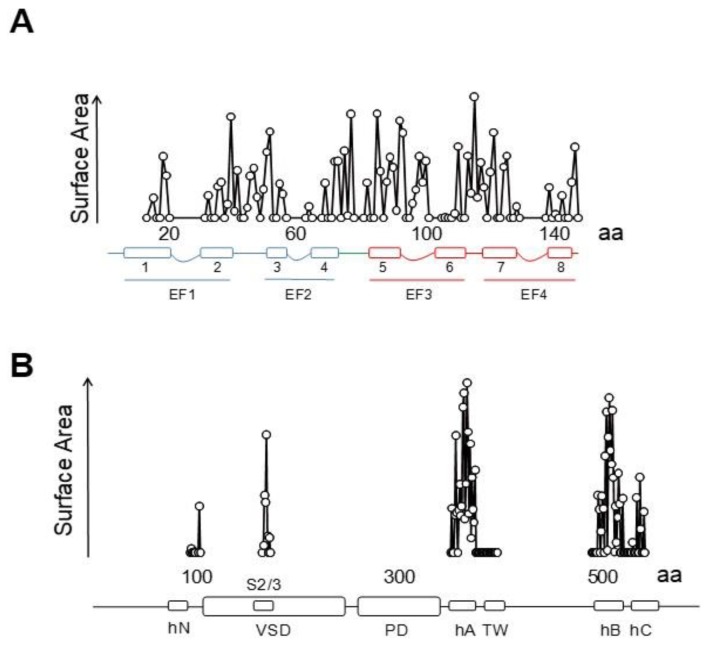
Plot of the contact surface per residue in CaM (**A**) and KCNQ1 (**B**). The cytosolic helix parallel to the membrane before S1 is labeled as hN. (Based on PDB 5VMS).

**Figure 16 ijms-21-01285-f016:**
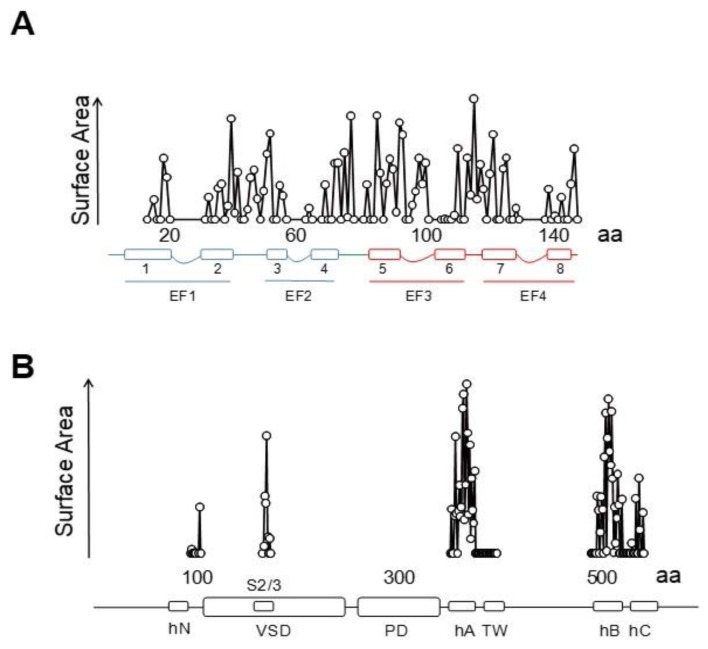
Hypothetical PIP_2_ binding site influenced by CaM EF3. The pore residues (GYG) were used as a reference to align the Kir structure (PDB 3SYA) to KCNQ1 (5VMS). The PIP_2_ molecule was manually moved maintaining the distance from the membrane to the presumed docking site, and manually adjusted to avoid obvious clashes. The docking space is delimited by pre-S1, S2/3 and S4/5 linkers. The positively charged Arg and Lys residues in KCNQ1 are represented as sticks using the CPK coloring scheme, as well as CaM Asn 98 and Tyr 100.

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
