# Peer review of "Atomistic Insights of Calmodulin Gating of Complete Ion Channels"

_ijms, 2020, doi:10.3390/ijms21041285_

Round 1

Reviewer 1 Report

The manuscript by Núñez et al. summarizes the recent understanding of the molecular mechanism of the effects of CaM in the gating of several 6TM channels based on mainly the structural data. The timely manuscript includes a lot of detailed analysis and description and will help readers to better understand the critical regulatory roles of CaM in these channels. I enjoyed reading the manuscript in general, and have the following suggestions for further improving it.

The major focus of this review is the K+ channel (as can been found in the abstract and the introduction). However, the authors also included TRPV5/V6 that are not K+ channels. I understand that the author included them to compare with the other K+ channel since their gating is also regulated by CaM and their structures with CaM are available. However, it will help if the authors can give a good reason to explain why TRPV5/6 channels are included. I found that the current 17 topics in this manuscript are difficult to follow due to the fact that the authors are discussing 4 different channels and they are all mixed in these 17 topics. To help the readers follow the flow, it will be helpful to add another layer of the subtitles to group the current topics 2-16 based on the channel type. For example, a subtitle can be added for the current topics 2-5 since they all talked by the Eag channel, while another subtitle can be added for current topics 6-10 for the SK channel. The same applied to TRPV5/V6 and KCNQ channels. I found references are missing in many placed throughout the manuscript. The authors should check the manuscript extensively and make sure always to cite properly. Below are some of the places that need references. I am sure it is far from a complete list:

All three paragraphs on page 2; line 87 (missing the reference for prokaryotic channel); line 92-94; lines 183-187; lines 244-246; lines 313-317; lines 326-333; lines 347-349; lines 372-374.

Line 96-98, why the result highlighted the importance of the interface between S1 within the VSD and S5 of PD? Can you explain further? Besides the cartoon structures of the channels in Fig. 1, 4, 9, and 12, showing the actual structures will help the readers have a better idea about the 3D structure. This also applied to some other places that only cartoons are shown. Although the method is cited with reference, a brief introduction of the method used in calculating the contact surface in Figures 2, 5, 10, and 15 will help the readers to understand the meaning of these figures. Line 175: Since this channel is called “Eag1” throughout the manuscript, “Kv10” should be changed here to be consistent. Line 181-183: The authors wrote, “However, some mutations lead to 15 fold increase in current [28], which may entail that more than 93% of the WT channels are closed at rest at a given moment”. However, this can only be true when these mutations only affect channel open probability, but not signal channel conductance. Lines 186-190: Although there is only a small increase of the current after bathing the patch in 0 Ca2+ solution, how can we tell that all binding CaM and the associated Ca2+ had been removed from the channel? Also, the last sentence has a grammar issue and needs to be rewritten. Line 192: what are the “Eag1 sites”? Line 202-203: I guess the “structure of the PAS domain in solution” and “20 structural NMR models” are talking about the hERG channel. If it is, please clarified in the sentence. Fig. 3A: It will be very helpful to use the same colors in the labeling as they are used for the domains. It is also the case for situations in the other similar figures. Lines 214-237: Since two channels, Eag1 and hERG, are described here, it is confusing at some places when the authors do not clarify which channel they are talking about. Line 244: “replace His 343 for Arg” should be “replace His 343 WITH Arg”. Lines 344-345: How the idea is challenged in the reports in 2014 and 2018? Why was it discarded? Line 448: The color of TRPV channels is purple instead of pink. Lines 461-463: the sentence “Since CK2 requires positively…” has grammar issues that need to be fixed. Line 499: “PCa/PNa 100>1” should be “PCa/PNa >100”. Line 567-569: The sentence “Interactions with…” has grammar issues that need to be fixed. Fig. 12A: The colors of the PD are switched. Line 690: The subject is missing in this sentence. Line 726: “then” should be “them.” Line 805-807: The result from the mutation only shows tetramerization affect both CaM association and the PIP2 sensitivity, but does not necessarily mean tetramerization affect the “interplay” between CaM and PIP2.

Author Response

We would like to thank the constructive criticism of the reviewers that has helped us to further improve the manuscript. We are glad that the paper was found to timely, useful, and enjoyable to read.

Reviewer 1

The major focus of this review is the K+ channel (as can been found in the abstract and the introduction). However, the authors also included TRPV5/V6 that are not K+ channels. I understand that the author included them to compare with the other K+ channel since their gating is also regulated by CaM and their structures with CaM are available. However, it will help if the authors can give a good reason to explain why TRPV5/6 channels are included.

The referee is right. The have added the following sentence (lines 74-76)

Although TRPV5/6 channels are not K+ selective, they will be discussed here because they have a similar 6TM architecture, their gating is regulated by CaM and their 3D structures are available.

I found that the current 17 topics in this manuscript are difficult to follow due to the fact that the authors are discussing 4 different channels and they are all mixed in these 17 topics. To help the readers follow the flow, it will be helpful to add another layer of the subtitles to group the current topics 2-16 based on the channel type. For example, a subtitle can be added for the current topics 2-5 since they all talked by the Eag channel, while another subtitle can be added for current topics 6-10 for the SK channel. The same applied to TRPV5/V6 and KCNQ channels.

Thank you for this suggestion. We have now organized the paper into four section, one for each type of channel. In addition, there is a section for the introduction and a “summary” and an “outlook”.

 I found references are missing in many placed throughout the manuscript. The authors should check the manuscript extensively and make sure always to cite properly. Below are some of the places that need references. I am sure it is far from a complete list: All three paragraphs on page 2; line 87 (missing the reference for prokaryotic channel); line 92-94; lines 183-187; lines 244-246; lines 313-317; lines 326-333; lines 347-349; lines 372-374.

Thank you very much for such a detailed response. We have included all the references. In addition, we have included references in the summary table. Thank you very much indeed.

 Line 96-98, why the result highlighted the importance of the interface between S1 within the VSD and S5 of PD? Can you explain further?”

To make this clearer, we have changes this sentence as follows (lines 98-99):

EagI and hERG (Kv10 and Kv11), S1 and S5 make extensive contacts, representing the main interface between the VSD and the PD. In these channels, the covalent linkage between S4 and S5 can be broken with little effect on the movement of S4 in response to voltage [7]. The resulting channel still gates the pore in response to voltage [14-16], highlighting the importance of the interface at the membrane

 Besides the cartoon structures of the channels in Fig. 1, 4, 9, and 12, showing the actual structures will help the readers have a better idea about the 3D structure. This also applied to some other places that only cartoons are shown.

We agree that the 3D images will help the reader. The 3D rendering of representative pdb files for each channel has been include, with lateral, to and bottom views in Figs. 1, 4, 8  and 12.

 Although the method is cited with reference, a brief introduction of the method used in calculating the contact surface in Figures 2, 5, 10, and 15 will help the readers to understand the meaning of these figures.

We have included the following sentence in legend of Figure 2 (lines 181-185):

“Contact surface area” between atoms A and B is defined as the area of the sphere whose center is the center of atom A and whose radius equals the sum of the Van der Waals radii or atom A and a solvent molecule. The solvent-accessible and contact surface of every atom has been computed using the software made available by Sobolev et al., http://oca.weizmann.ac.il/oca-bin/lpccsu [27].

 Line 175: Since this channel is called “Eag1” throughout the manuscript, “Kv10” should be changed here to be consistent.

Thank you very much. It has been amended accordingly.

 Line 181-183: The authors wrote, “However, some mutations lead to 15 fold increase in current [28], which may entail that more than 93% of the WT channels are closed at rest at a given moment”. However, this can only be true when these mutations only affect channel open probability, but not signal channel conductance.

The referee is completely right, and we appreciate the opportunity to amend this over-interpretation of the data. The sentence has been amended, and now it reads:

However, only a slight increase in current density was observed, which is lower than expected if most channels were inhibited by CaM at rest and all Ca2+ was effectively removed. This remarkable report shows a slight increase of the current after patch excision into a medium without Ca2+. When the patch was bathed with 200 nM Ca2+ with CaM, the current was suppressed. After bathing in 0 Ca2+, current levels returned to the on-cell patch levels [25].

 Lines 186-190: Although there is only a small increase of the current after bathing the patch in 0 Ca2+ solution, how can we tell that all binding CaM and the associated Ca2+ had been removed from the channel? Also, the last sentence has a grammar issue and needs to be rewritten.

We assumed that calcium was absent, as claimed by the author. Nevertheless, we have removed this sentence that may add confusion to the paper.

 Line 192: what are the “Eag1 sites”?

We clarify this in line 203: GST proteins fused to Eag1 N1 and C2 sequences bind CaM…

 Line 202-203: I guess the “structure of the PAS domain in solution” and “20 structural NMR models” are talking about the hERG channel. If it is, please clarified in the sentence.

The referee is right again. We have amended the text (line 215): …none of the 20 structural NMR models for hERG reflect the position of the PAS-Cap…

 Fig. 3A: It will be very helpful to use the same colors in the labeling as they are used for the domains. It is also the case for situations in the other similar figures.

Thank you very much for the suggestion.

 “Lines 214-237: Since two channels, Eag1 and hERG, are described here, it is confusing at some places when the authors do not clarify which channel they are talking about.”

Asfd

 Line 244: “replace His 343 for Arg” should be “replace His 343 WITH Arg”.

Thank you very much.

Lines 344-345: How the idea is challenged in the reports in 2014 and 2018? Why was it discarded?

This has been clarified as follows (lines 359-361): …because no evidence was found for the existence of 2:2 complexes, and it was proposed that CaM was crosslinking adjacent subunits [35; 52], but it prevailed until it was definitely discarded in 2018 based on cryo-EM images [45]

Line 448: The color of TRPV channels is purple instead of pink.”

Amended. Thank you very much.

Lines 461-463: the sentence “Since CK2 requires positively…” has grammar issues that need to be fixed.

Lines 461-463, now 479-481, have been amended:

CK2 requires positively charged compounds to phosphorylate CaM [64], but these compounds are not present in functional excised patch experiments. As an alternative, it has been suggested that the charged residues at the base of S1 serve this purpose [63].

 Line 499: “PCa/PNa 100>1” should be “PCa/PNa >100”.

Amended. Thank you very much.

 Line 567-569: The sentence “Interactions with…” has grammar issues that need to be fixed.

Lines 567-569, now 589-590 have been amended: Interactions with residues downstream of Trp 583 also contribute to the binding site, where the second most important residue is Gln 587 for TRPV5 and His 587 for TRPV6 (Fig. 10).

 Fig. 12A: The colors of the PD are switched.

Actually, the colouring was on purpose to convey the swapped configuration of the channel… but we agree that is confusing. We have connected the VSD to its corresponding PD domain with dashed lines to prevent misinterpretations of the figure.

Line 690: The subject is missing in this sentence.

Amended. Thank you very much.

Line 726: “then” should be “them.”

Amended. Thank you very much.

Line 805-807: The result from the mutation only shows tetramerization affect both CaM association and the PIP2 sensitivity, but does not necessarily mean tetramerization affect the “interplay” between CaM and PIP2.

This has been amended. Thank you very much (line 832): Both CaM and PIP2 dependent regulation are affected by the tetramerization domain

Reviewer 2 Report

I am an expert in neither calmodulin nor in 6TM K channels. I read the review “Atomistic insights of calmodulin gating of complete ion channels” by Nunes and coworkers with real difficulty. I doubt that this work is currently in a publishable form.

I do not see it as a proper review providing the necessary broad background or the relevance of the molecular information to physiology. It at least ought to provide a minimal history of the topic and the key observations that led to today’s discussion of structures. Instead, this manuscript is largely a poorly systematized compendium of raw facts taken from original papers and supplemented with schematic graphical illustrations. The biophysics that might be behind the positive or negative channel regulation has not been adequately presented even in the form of a hypothesis.

However, the work might be interesting to a broader audience because it describes a number of non-canonical modes of Cam binding to its diverse targets. The work also focuses on subtle physiological roles of several types of K and cationic channels, mutations in which have been linked to pathologies.

Here are my suggestions to make this manuscript easily readable and more attractive to a broader audience:

Introduce Cam signaling and outline the major physiological processes that clearly imply the involvement of Cam in shaping action potentials in different settings (neurons, heart, smooth muscle, etc.). Illustrate the ‘classical’ modes of Cam binding to targets and the more recently discovered non-canonical ways of symmetric and asymmetric (one-lobed) interactions. Illustrate the compact and stretched modes of binding by presenting fragments of real structures. Outline the four families of K channels for which interactions with Cam have been shown. Describe the importance of the cross-talk between repolarization (and resting potential regulation) with intracellular Ca2+ concentration. Provide adequate descriptions of the normal physiological processes for which these channels are critical, and inform the reader about the anticipated or established effects of Cam on these channels. Clearly outline the functional cycle of each channel, the ability to inactivate, and the preferential state it occupies. Explain that the Ca-Cam effect can be both activation and silencing of the channel. In the parts devoted to a specific family of channels, avoid detailed and convoluted lines describing previously discarded hypotheses. Instead, try to present mechanistic views for the interpretation of the structural information. The manuscript should be professionally edited.

Author Response

We would like to thank the constructive criticism of the reviewers that has helped us to further improve the manuscript. We are glad that the paper was found to timely, useful, and enjoyable to read.

We appreciate the effort that the referee has made to asses our manuscript, considering that he/she admits not being conversant with ion channels or calcium-calmodulin signalling, and had difficulties to follow this concise collection of information, hypothesis and concepts used for a critical appraisal of the state of the art in the field. He/she proposes to extend and transform the manuscript into reviews of overlapping topics. We respectfully find that his/her proposal is inadequate because, among other things, there are already several reviews, many cited in our manuscript, which address those topics.

Reviewer 3 Report

II am glad the authors wrote this review. It is a well-written, needed, and useful summary of the status of calmodulin gating and how this gating is governed by different mechanisms in four types of tetrameric ion channels.

The architecture of the review is well constructed, the illustrations are adequate, and the final table is appreciable.
Only downside for this last table, the references are missing and should appear there.

Author Response

We would like to thank the constructive criticism of the reviewers that has helped us to further improve the manuscript. We are glad that the paper was found to timely, useful, and enjoyable to read.

I am glad the authors wrote this review. It is a well-written, needed, and useful summary of the status of calmodulin gating and how this gating is governed by different mechanisms in four types of tetrameric ion channels.

The architecture of the review is well constructed, the illustrations are adequate, and the final table is appreciable. Only downside for this last table, the references are missing and should appear there.

Thank you for your comments. The references have been added to the table.

Round 2

Reviewer 2 Report

Nunes and coworkers explore a fascinating topic in their review. Ca2+ sensing by 6TM channels mediated by calmodulin is outlined here through a set of recently solved structures, and the authors explore these structures by mapping atomic contacts (buried molecular area) between these channels and calcified calmodulin, with no comparison to other systems. Some of the structures are solved under high Ca2+ and their nativity can be questioned.

With the exception of maybe 20-50 people who work in the field, the general reader will likely be deterred by the lack of physiological and the cellular context in which these structures are analyzed and interpreted. It is not even clearly denoted which CALM (1, 2 or 3) and which isoform is engaged in each of the interactions.

My suggestion is to modify the abstract such that right at the outset the reader would expect what the take-home message is. Somewhere around lines 106, and 297 the authors should expand the description of the physiological context where the CALM-mediated modulation of respective channel activities takes place. 

Author Response

Dear Editor,

We would like to thank the Reviewer 2 for the comments and suggestions. We have taken into account her/him considerations and the manuscript has been modified.

Reviewer 2

Nunes and coworkers explore a fascinating topic in their review. Ca2+ sensing by 6TM channels mediated by calmodulin is outlined here through a set of recently solved structures, and the authors explore these structures by mapping atomic contacts (buried molecular area) between these channels and calcified calmodulin, with no comparison to other systems. Some of the structures are solved under high Ca2+ and their nativity can be questioned.

My suggestion is to modify the abstract such that right at the outset the reader would expect what the take-home message is.

Thank you very much for the suggestion. The abstract has been modified as follows:

In combination with mutagenesis-function, structural information of individual domains and functional studies, different mechanisms employed by CaM to control channel gating are starting to be understood at atomic detail. Although each CaM lobe engages through apparently similar alpha-helices, they do so using different docking strategies which allow selective action of drugs with great therapeutic potential. Here, new insights regarding four types of tetrameric channels with six transmembrane (6TM) architecture, Eag1, SK2/SK4, TRPV5/TRPV6 and KCNQ1-5, and its regulation by CaM are described structurally. Different CaM regions, N-lobe, C-lobe and EF3/EF4-linker play prominent signaling roles in different complexes, emerging the realization of crucial non-canonical interactions between CaM and its target that are only evidenced in the full-channel structure. Different mechanisms to control gating are used, including direct and indirect mechanical actuation over the pore, allosteric control, indirect effect through lipid binding, as well as direct plugging of the pore. Although each CaM lobe engages through apparently similar alpha-helices, they do so using different docking strategies. We discuss how this allows selective action of drugs with great therapeutic potential.

It is not even clearly denoted which CALM (1, 2 or 3) and which isoform is engaged in each of the interactions.

To clarify that all CALM genes encode an identical protein, the introduction has been modified by including the following paragraph (line 38):

CaM is formed by two similar globular domains, the N- and C-lobes linked by a very flexible sequence. Each lobe is composed of two EF-hands which are responsible for binding of up to four Ca2+ ions. CaM targets are usually amphipathic helical protein regions rich in hydrophobic and basic residues. CaM lobes can be in an open, semi-open or closed configuration depending on Ca2+ occupancy. In addition, the abundance of methionine residues confers another level of plasticity at the amino acid level. These characteristics enable CaM to bind to more than 300 targets with little sequence similarity. The fact that there are three genes in humans (CALM1-3) which encode an identical CaM protein emphasizes its critical role in physiology [1]. Here, we review several high resolution structures of CaM in complex with ion channels, which provide an essential framework to understand CaM-mediated regulation. Within the channels discussed here, Ca2+-CaM inhibits Eag1 (Kv10) and TRPV5/6 channels, activates SK channels, and inhibits or activates Kv7 channels depending on other factors.

Somewhere around lines 106, and 297 the authors should expand the description of the physiological context where the CALM-mediated modulation of respective channel activities takes place. 

         Thank you for your suggestion, we have partly addressed this concern in the previous paragraph. In addition, we have added the following sentence (lines 337-338):

At resting Ca2+ levels, CaM is bound to the closed channel and subsequent Ca2+ elevation leads to pore opening.